# Estimating the State of a Geophysical System with Sparse Observations : Time Delay Methods to Achieve Accurate Initial States for Prediction

Zhe An[1], Daniel Rey[1], JingXin Ye[1], and Henry D. I. Abarbanel[1, 2]

[1]Department of Physics, University of California, San Diego, 9500 Gilman Drive, La Jolla, CA 92093-0374, USA
[2]Marine Physical Laboratory (Scripps Institution of Oceanography) University of California, San Diego 9500 Gilman Drive La Jolla, CA 92093-0374, USA

*Correspondence to:* Zhe An (z1an@ucsd.edu )

**Abstract.** The problem of forecasting the behavior of a complex dynamical system through analysis of observational time-series data becomes difficult when the system expresses chaotic behavior and the measurements are sparse, in both space and/or time. Despite the fact that this situation is quite typical across many fields, including numerical weather prediction, the issue of whether the available observations are 'sufficient' for generating successful forecasts is still not well-understood. An analysis by Whartenby *et al.* (2013) found that in the context of the nonlinear shallow water equations on a $\beta$-plane, standard nudging techniques require observing approximately 70% of the full set of state variables. Here we examine the same system using a method introduced by Rey *et al.* (2014a), which generalizes standard nudging methods to utilize time delayed measurements. We show that in certain circumstances, it provides a sizable reduction in the number of observations required to construct accurate estimates and high-quality predictions. In particular, we find that this estimate of 70% can be reduced to about 33% using time delays, and even further if Lagrangian drifter locations are also used as measurements.

## 1 Introduction

The ability to forecast the complex behavior of global circulation in the coupled Earth system lies at the core of modern numerical weather prediction (NWP) efforts. To successfully predict such behavior requires both a good model of the underlying physical processes as well as an accurate estimate of the state of the model at the end of the analysis or observation window. When the model is chaotic, even if it is known precisely, the accuracy of the prediction depends on the accuracy of the initial state estimate. This is due to sensitive dependence to the initial conditions, which was first identified by Lorenz (1963).

Here we consider an idealized situation where a perfect dynamical model describes the deterministic time evolution of a set of $D$ state variables. We assume that $L$ measurements are made at uniform time intervals $\Delta t$ within an observation window of length $T$, so the total number of distinct measurements is $L \times (T/\Delta t + 1)$. Our main concern here is the case where the measurements are sparse in state space, so $(L \ll D)$.

This situation of high dimensional dynamics and sparse measurements is typical in the process of examining the consistency of observed data and quantitative models of complex nonlinear systems: from functional nervous systems to genetic transcrip-

tion dynamics, among many other examples (Abarbanel, 2013). Although the methods described here have broad applicability across the quantitative study of the underlying physical or biological properties appearing in many complex systems, our discussion will focus solely on a specific geophysical system: the shallow water equations. As discussed by Cardinali (2013), operational NWP models at the European Centre for Medium-Range Weather Forecasting (ECMWF) now contain upwards of $10^8$ degrees of freedom. These models are analyzed using $3 - 4 \times 10^7$ daily observations, a large portion of which are often discarded. Given the scale of these calculations, the question of whether the remaining observations are in fact 'sufficient' for producing accurate analyses and forecasts is of considerable importance.

To clarify these ideas we refer to the observability study given by Whartenby *et al.* (2013), which evaluated the performance of familiar nudging methods on chaotic, shallow water flow. The flow was simulated on a $\beta$-plane defined by a square grid with uniform spacing $N_\Delta$, periodic boundary conditions, and driven by Ekman pumping. Poor predictions were obtained unless the height variable $h$ and at least one of the two velocity variables $u, v$ at each of the $N_\Delta \times N_\Delta$ grid points were measured. In other words, accurate forecasts required direct observation of roughly 70% of the $3N_\Delta^2$ dynamical variables.

This lower bound was termed the critical number of measurements $L_s$ required to synchronize the model with the data. Synchronization occurs when the error between the model state estimate and the data drops below a prescribed threshold, which is typically below the magnitude of the observation noise. It depends on a number of factors, including the type of observation network, the signal to noise ratio, properties of the model such as the number and magnitude of its Lyapunov exponents, as well as the choice of algorithm. Strong constraint 4DVar for instance, which is now standard practice in data assimilation Rabier *et al.* (2000), encounters serious difficulty when the length of the window is long relative to the timescale of the chaos (Pires *et al.*, 1996). In this case, the algorithm will not produce adequate forecasts even with full observations $L = D$. Despite this however, the lowest estimates of $L_s$ appear remarkably consistent between nudging methods and fixed interval formulations of 4DVar with both strong and weak constraints (Abarbanel *et al.*, 2009; Abarbanel, 2013; Quinn & Abarbanel, 2010).

Here we examine what can be done when $L < L_s$. Specifically, we will show that using the method introduced by Rey *et al.* (2014a, b), which modifies a standard nudging technique to include additional information in the *time delays* of the observations, the estimate of 70% given by Whartenby *et al.* (2013) can be reduced to roughly 33%, and can be even further reduced if positional observations from Lagrangian drifters are also used. These outcomes suggest that time delays may be useful for reducing the number of required observations to meet the practical constraints of operational NWP. However, further testing with more realistic models, observations, and noise are required to verify this claim.

## 2   Time delayed nudging

We now briefly discuss the concept of time delayed nudging, further details can be found in Rey *et al.* (2014a, b). The system is assumed to be described by a mathematical model, whose state is given by a $D$-dimensional vector $\mathbf{x}(t)$. The model defines a dynamical rule for evolving the $\mathbf{x}(t)$ in time, which we assume can be represented as a set of ordinary differential equations

(ODEs)

$$\frac{d\mathbf{x}(t)}{dt} = \mathbf{F}(\mathbf{x}(t), t). \tag{1}$$

If the dynamics of the system are given by partial differential equations (PDEs), such as with fluids in an earth systems model, these ODEs may be realized by discretizing the PDEs on a grid. It is worth noting however, that a non-trivial amount of discretization error is introduced in this process.

Measurements of the physical system are recorded during an observation window $0 \leq t \leq T = N \Delta t$, where $L$ observations $\mathbf{y}(t_n)$ are taken at each time $t_n = n \Delta t$ for $n = 0, 1, \ldots, N$. The measurements $\mathbf{y}(t)$ are related to the state vector $\mathbf{x}(t)$ through a measurement operator $\mathbf{h}(t)$. For simplicity, it is taken here to be a constant, $L \times D$ projection matrix $\mathbf{h}(t) = \mathbf{H}$, so that $\mathbf{y}(t) = \mathbf{H} \cdot \mathbf{x}(t) + \text{noise}$. The total number of measurements in the observation window is $L \times (N+1)$.

The overall objective is to estimate the full model state $\mathbf{x}(T)$ at the end of the assimilation window using information from observations, and then use this estimate to predict the system's subsequent behavior for $t > T$ using Eq. (1). The accuracy of these predictions, when compared with additional measured data in the prediction window $t > T$, serves as a metric to validate both the model and the assimilation method, through which the unobserved states of the system are determined. This establishes a necessary condition on $L$ that is required to synchronize the model output with the data and thereby obtain accurate estimates for the unobserved states of the system.

When the model is known precisely, a familiar strategy for transferring information from the measurements to the model involves the addition of a coupling or control or nudging term to Eq. (1),

$$\frac{d\mathbf{x}(t)}{dt} = \mathbf{F}(\mathbf{x}(t), t) + \mathbf{H}^{\dagger} \cdot \mathbf{G}(t) \cdot \big(\mathbf{y}(t) - \mathbf{H} \cdot \mathbf{x}(t)\big). \tag{2}$$

where $\mathbf{H}^{\dagger}$ denotes the transpose, and $\mathbf{G}(t)$ is an $L \times L$ matrix that is nonzero only at times $t_n$ where measurements occur. For simplicity, when $\mathbf{G}(t)$ is non-zero, it is assumed to be constant and diagonal, so the coupling terms only affect the measured states.

This long-standing procedure, known as 'nudging' in the geophysics and meteorology literature, has been shown to fail when the number of measurements at a given time falls below a critical value $L_s$ (Abarbanel *et al.*, 2009). This can be understood by noting that the coupling term perturbs the observed model states, driving them towards the data. With enough observations $L$, and a sufficiently strong coupling $\mathbf{G}(t)$, this control term alters the Jacobian of the dynamical system Eq. (2) so that all its (conditional) Lyapunov exponents are negative (Pecora & Carroll, 1990; Abarbanel, 1996; Kantz & Schreiber, 2004). That is, the log of the maximum eigenvalue of the matrix $[\mathbf{\Phi}(T, t_0)^{\dagger} \cdot \mathbf{\Phi}(T, t_0)]^{1/2T}$ is negative, where $\mathbf{\Phi}(t, t') = \partial \mathbf{x}(t) / \partial \mathbf{x}(t')$ is the linearized state transition matrix or tangent linear model. Its time evolution is described by the variational equation

$$\frac{d\mathbf{\Phi}(t, t')}{dt} = \mathbf{D}\tilde{\mathbf{F}}(\mathbf{x}(t), t) \cdot \mathbf{\Phi}(t, t') \qquad \Phi_{ab}(t, t') = \delta_{ab}, \tag{3}$$

along the trajectory given by Eq. (2) and

$$\mathbf{D}\tilde{\mathbf{F}} = \mathbf{DF}(\mathbf{x}(t), t) - \mathbf{H}^{\dagger} \cdot \mathbf{G}(t) \cdot \mathbf{H}$$

is its Jacobian. That is, $DF_{ab}(\mathbf{x}(t),t) = \partial F_a(\mathbf{x}(t),t)/\partial x_b(t)$ and $\delta_{ab}$ is the Kronecker delta, so $\Phi(t',t')$ is an identity matrix. This establishes a necessary condition on $L$ required to synchronize the model with the data. Numerical experiments have shown that when this condition is not met, estimates are not accurate and predictions are unreliable (Abarbanel *et al.*, 2009; Abarbanel, 2013). An example of this will be given later in our discussion of geophysical shallow water flow.

It is therefore important to understand for a given problem, whether $L > L_s$. If this condition is not satisfied and additional measurements cannot be made, then we must find another means to overcome this deficit in $L$. One way to proceed involves the recognition that additional information resides in the temporal derivatives of the observations. In practice, however, this derivative information cannot be measured directly, although it can be approximated via finite differences, for instance by approximating $d\mathbf{y}(t_n)/dt$ with $(\mathbf{y}(t_n + \tau) - \mathbf{y}(t_n))/\tau$ where $\tau$ is some multiple of the time differences between measurements. The drawback here is that the derivative operation acts as a high-pass filter, and is thus quite susceptible to noise in the measurements. Alternatively, it has been known for some time in the nonlinear dynamics literature that this additional information in the derivative is also available in the *time delay* of the measurements, $\mathbf{y}(t_n + \tau)$. This process can be repeated as many times as needed to form a $D_M$ dimensional vector of time delays, which we call $\mathbf{S}(t)$.

This idea provides the basis for the well-established technique in the analysis of nonlinear dynamical systems, where this structure is employed as a means of reconstructing unambiguous orbits of a partially observable system (see e.g., Aeyels (1981a, b); Mañé (1981); Sauer *et al.* (1991); Takens (1981); Kantz & Schreiber (2004); Abarbanel (1996)). By mapping to a proxy space of time delayed observations, one is able invert the projection associated with measuring $L < D$ components of the underlying dynamics, by using fact that new information beyond $\mathbf{y}(t_n)$ lies in $\mathbf{y}(t_n + \tau)$. The derivative operation is just another (albeit less numerically robust) way of accessing this information.

Note that the *time delay* $\tau$ and the *embedding dimension* $D_M$ are parameters that need to be chosen appropriately for the system, although a number of useful heuristics are available (Abarbanel, 1996). Moreover, Takens (1981) proved that that taking $D_M > 2 D_A$, where $D_A$ is the fractal dimension of the attractor, is sufficient to unambiguously reconstruct the topology of the attractor. It is worth noting however that this condition is only sufficient, and the procedure often succeeds with considerably a smaller value of $D_M$.

In the estimation context, the time delays are used in a slightly different way. Instead of reconstructing the topology of the attractor, they are used to control local instabilities in the dynamics, which cause errors in the analysis to grow. In other words, $D_M$ does not need to embed the entire space. Rather, it only needs to be large enough to effectively increase the amount of information transferred from the $L$ measurements to a value above the critical threshold, $L_s$.

Using this idea Rey *et al.* (2014a, b) proposed a technique to extract additional information from time delayed observations by constructing an extended state space $\mathbf{S}(t)$, created from an $L \cdot D_M$ dimensional vector of the measurements and its time delays. The components of this time delayed observation vector are denoted by

$$\mathbf{Y}^\dagger(t_n) = \{\mathbf{y}^\dagger(t_n), \mathbf{y}^\dagger(t_n + \tau), \dots, \mathbf{y}^\dagger(t_n + \tau(D_m - 1))\}, \tag{4}$$

where $D_M$ is the dimension of the time delayed vector $\mathbf{Y}(t_n)$, and $\tau$ is the delay, which here is assumed to be a positive integer multiple of $\Delta t$. Also, note that the term 'delay' here is not used in its usual sense. Rather, this method involves a time *advanced* formulation, which for positive $\tau$ incorporates observations at later times. Both formulations are acceptable however.

The corresponding time delay model vectors $\mathbf{S}(\mathbf{x}(t))$ are given by

$$\mathbf{S}^\dagger(\mathbf{x}(t)) = \{[\mathbf{H} \cdot \mathbf{x}(t)]^\dagger, [\mathbf{H} \cdot \mathbf{x}(t+\tau)]^\dagger, \dots, [\mathbf{H} \cdot \mathbf{x}(t+\tau(D_m-1))]^\dagger\}, \tag{5}$$

where the values $\mathbf{x}(t' > t)$ are constructed by integrating the *uncoupled* dynamics, Eq. (1), forward in time. The time evolution for $\mathbf{S}(\mathbf{x}(t))$ is given by the chain rule,

$$\frac{d\mathbf{S}(\mathbf{x}(t))}{dt} = \mathbf{DS}(\mathbf{x}(t)) \cdot \mathbf{F}(\mathbf{x}(t),t), \tag{6}$$

where the Jacobian $\mathbf{DS}(\mathbf{x}(t)) = \partial\mathbf{S}(\mathbf{x}(t))/\partial\mathbf{x}(t)$ with respect to $\mathbf{x}(t)$ can be computed using the variational Eq. (3), by substituting the Jacobian of the uncoupled model $\mathbf{D\tilde{F}} \to \mathbf{DF}$. Furthermore, in analogy with Eq. (2), we introduce a control term

$\mathbf{g}(t)$ in time delay space

$$\frac{d\mathbf{S}(\mathbf{x}(t))}{dt} = \mathbf{DS}(\mathbf{x}(t)) \cdot \mathbf{F}(\mathbf{x}(t),t) + \mathbf{g}(t) \cdot \big(\mathbf{Y}(t) - \mathbf{S}(\mathbf{x}(t))\big). \tag{7}$$

We then transform back to physical space, by multiplying both sides of this equation by $[\mathbf{DS}(\mathbf{x}(t))]^{-1}$, to get

$$\frac{d\mathbf{x}(t)}{dt} = \mathbf{F}(\mathbf{x}(t),t) + \mathbf{G}(t) \cdot [\mathbf{DS}(\mathbf{x}(t))]^{-1} \cdot \mathbf{g}(t) \cdot \big(\mathbf{Y}(t) - \mathbf{S}(\mathbf{x}(t))\big), \tag{8}$$

Note there are now two control terms, $\mathbf{G}(t)$ and $\mathbf{g}(t)$, which act in physical and in time delay space respectively. Also, since

$\mathbf{DS}(\mathbf{x}(t))$ is a $(L \cdot D_m) \times D$ matrix, it is generally not square so its pseudoinverse $[\mathbf{DS}(\mathbf{x}(t))]^+$ is used.

At each step of the integration of the controlled (nudged) dynamical equations Eq. (8), the control term perturbs the full state vector in time delay space $\mathbf{S}(\mathbf{x}(t))$ toward the time delay measurement vector $\mathbf{Y}(t)$, allowing it to extract additional information from the waveform of the *existing measurements*. The value of this statement will become more clear later on.

Furthermore, in the limit $D_M = 1$ the time delay formulation Eq. (8) reduces to the standard nudging control Eq. (2).

Two important differences however are realized when $D_M > 1$. First, information from the time delays of the observations is presented to the physical model equations. And second, all components of the model state $\mathbf{x}(t)$ are influenced by the control term, not just the observed components. This, for example, allows fixed parameters $\mathbf{p}$ of the model may be estimated as a natural result of the synchronization process by including them as additional state variables, satisfying $d\mathbf{p}(t)/dt = 0$.

Time delay nudging shares considerable overlap with incremental formulations of strong constraint 4DVar (Lewis *et al.*,

1985; Talagrand *et al.*, 1987; Courtier *et al.*, 1994). For instance, both methods use a sliding window of observations, and compute the control (nudging) perturbation by minimizing the squared magnitude of time-distributed innovations $|\mathbf{Y}(t) - \mathbf{S}(\mathbf{x}(t))|^2$. The use of time advanced observations is also standard practice in strong constraint 4DVar, and is motivated by the fact that the necessary conditions for synchronization require one to control the propagation of errors on the unstable manifold(Trevisan *et al.*, 2010; Palatella *et al.*, 2013). Since these errors are locally described by Eq. (3) as the system evolves

forward in time, the time advanced construction is a natural choice.

The main differences between the two methods are as follows:

1. Strong constraint 4DVar does not include the notion of a time delay or embedding dimension.

2. With the time delay method, the analysis is propagated in small increments $dt$ between analyses, and observations are re-used.

3. Time delay nudging uses truncated singular-value decomposition to regularize the solution, while strong constraint 4DVar uses a background term to perform Tikhonov regularization.

Regarding the second point, near the end of the observation window one must either switch to a time delayed formulation or reduce $D_M$ appropriately. Here however for simplicity, the end of the observation window is taken so that the last observation $\mathbf{y}(T + \tau(D_M - 1))$ is always available. Also, the third point prevents time delay nudging from being applied to systems of the size used in operational NWP. However, a simplified variation of time delay nudging was recently given by Pazo *et al.* (2016). This method avoids the variational Eq. (3) and the generalized inverse altogether, and thus requires considerably less computational effort than either time delay nudging or strong constraint 4DVar. It is worth investigating whether this method is also capable of achieving the same reduction in $L_s$, shown here for geophysical flows.

Furthermore, while we are currently working on unifying the motivating ideas behind time delay nudging with the variational action principle of weak constraint 4DVar, these and other related connections to 4DVar will be given in a subsequent paper. For the moment however, no additional theory will be introduced. Instead, we focus its application to a core geophysical model: the shallow water equations.

## 3 Twin Experiments

We test our time delay nudging procedure through a series of *twin experiments* (Durand *et al.*, 2002; Blum *et al.*, 2009; Blum, 2010). After solving the original dynamical equations Eq. (1) forward from preselected initial conditions $\mathbf{x}(0)$, the observation process is simulated by applying the observation operator $\mathbf{H}$ to project the state down to the $L$ observed components. Gaussian noise $N(0, \sigma)$ is then added to each component to simulate observation error. The estimation process continues as described above until the time $t = T$. At this point, the coupling terms $\mathbf{g}(t)$ and $\mathbf{G}(t)$ are set to zero and the uncoupled dynamics Eq. (1) are integrated forward from the estimated $\mathbf{x}(T)$ to construct a forecast for $t > T$. Comparing this forecast against additional observations $\mathbf{y}(t > T)$ then tests whether the unobserved states are also accurately estimated.

To simulate the conditions of a *true experiment* we monitor our progress by calculating the observable synchronization error, namely the root mean square deviation between the data and the observed model states

$$SE(t_n) = \sqrt{\frac{1}{L} |\mathbf{H} \cdot \mathbf{x}^s(t_n) - \mathbf{y}^s(t_n)|^2}. \tag{9}$$

In this expression, scaled variables have been introduced such that $x_\ell^s(t) = [x_\ell(t) - x_\ell^{min}(t)]/[x_\ell^{max}(t) - x_\ell^{min}(t)]$ and $x_\ell^{min/max}(t)$ are the minimum or maximum values of $x_\ell(t)$ over the entire assimilation window. The same definition holds for $y_\ell^s(t)$. This rescales all data and observed model states to lie in the interval $[0, 1]$, so that each state component's contribution to the syn-

chronization error is roughly equal. While this could make the result sensitive to outliers in the data, it did not appear to be an issue here.

It was previously shown by Whartenby *et al.* (2013) that when the synchronization error Eq. (9) decreases to very small values, the full state $\mathbf{x}(T)$ is accurately estimated and the forecast is quite good. Conversely, when this fails to occur, the full state $\mathbf{x}(T)$ is not well estimated and the prediction is unreliable. In Rey *et al.* (2014a, b), this contraction of the synchronization error was only observed when the number of time delayed observations $L \times D_M$, and the magnitude of the coupling matrices $\mathbf{g}(t)$, $\mathbf{G}(t)$ were 'large enough'. The precise meaning of this statement will become apparent shortly.

## 4 Nonlinear Shallow Water Equations

We now describe the application of time delay nudging to a nonlinear model of shallow water flow on a mid-latitude $\beta$-plane. This geophysical fluid dynamical model was previously examined by Pedlosky (1987) and Whartenby *et al.* (2013) as well as many others, and is at the core of earth system flows used in NWP. Of course, operational models contain considerably more detail than this example, and those models also describe the dynamics over a sphere. While we suspect the results presented here for this simplified model will be applicable to more realistic models as well, additional experiments are needed to validate this claim.

As the depth of the coupled atmosphere ocean fluid layer ($10 - 15$ km) is markedly less than the earth's radius (6400 km), the shallow water equations for two dimensional flow provide a good approximation to the fluid dynamics of the ocean. Three fields on a mid-latitude plane describe the fluid flow $\{u(\mathbf{r},t), v(\mathbf{r},t), h(\mathbf{r},t)\}$: the north-south velocity $v(\mathbf{r},t)$, the east-west velocity $u(\mathbf{r},t)$, and the height of the fluid $h(\mathbf{r},t)$, with $\mathbf{r} = \{x,y\}$. The fluid is taken as a single, constant density layer and is driven by wind stress $\tau(\mathbf{r},t)$ at the surface $z = h(\mathbf{r},t)$ through an Ekman layer. These physical processes satisfy the following dynamical equations with $\mathbf{u}(\mathbf{r},t) = \{u(\mathbf{r},t), v(\mathbf{r},t)\}$,

$$\frac{\partial \mathbf{u}(\mathbf{r},t)}{\partial t} = -\mathbf{u}(\mathbf{r},t) \cdot \nabla \mathbf{u}(\mathbf{r},t) - g \nabla h(\mathbf{r},t) + \mathbf{u}(r,t) \times (f(y)\hat{\mathbf{z}}) + A \nabla^2 \mathbf{u}(\mathbf{r},t) - \epsilon \mathbf{u}(\mathbf{r},t)$$

$$\frac{\partial h(\mathbf{r},t)}{\partial t} = -\nabla \cdot [h(\mathbf{r},t) u(\mathbf{r},t)] - \hat{z} \cdot \mathrm{curl} \left[ \frac{\tau(\mathbf{r},t)}{f(y)} \right]. \tag{10}$$

The Coriolis force is linearized about the equator $f(y) = f_0 + \beta y$ and the wind-stress profile is selected to be $\tau(\mathbf{r},t) = \{[F/\rho] \cos(2\pi y), 0\}$. The parameter $A$ represents the viscosity in the shallow water layer, $\epsilon$ is Rayleigh friction and $\hat{z}$ is the unit vector in the z-direction. The values we have used for the model parameters are given in Table 1. With these fixed parameters the shallow water flow is chaotic, and the largest Lyapunov exponent for this flow is estimated to be $\lambda_{max} = 0.0325/h \approx 1/31h$ by measuring the average growth rate of random perturbations. The details of this calculation are given by Whartenby *et al.* (2013).

We have analyzed this flow using the enstrophy conserving discretization scheme given by Sadourny (1975) on a grid of size $N_\Delta^2$ for increasing resolution $N_\Delta = \{16, 32, 64\}$. The total domain size is constant $800 \times 800$ km and periodic boundary conditions are enforced. Using the twin-experiment framework, with simple nudging given in Eq. (2) and a static, uniform observation operator, approximately $70\%$ of the $D = 3N_\Delta^2$ degrees of freedom were required observed to synchronize the

| Parameter | Physical Quantity | Value in Twin Experiments |
|:---:|:---:|:---:|
| $\Delta t$ | Time Step | 36 s |
| $\Delta X$ | East-West Grid Spacing | 50 km |
| $\Delta Y$ | North-South Grid Spacing | 50 km |
| $H_0$ | Equilibrium Depth | 5.1 km |
| $\varphi_0$ | Central latitude of the $\beta$ plane | $3.6°$ |
| $f_0$ | Central value of the Coriolis parameter | $5 \times 10^{-5}$ s$^{-1}$ |
| $\beta$ | Meridional derivative of the Coriolis parameter | $2.0 \times 10^{-11}$m$^{-1}$s$^{-1}$ |
| $F/\rho$ | Wind Stress | $0.2$ m$^2$s$^{-3}$ |
| $A$ | Effective Viscosity | $10^{-4}$ m$^2$s$^{-1}$ |
| $\epsilon$ | Rayleigh Friction | $2 \times 10^{-8}$ s$^{-1}$ |

**Table 1.** Parameters used in the generation of the shallow water 'data' for the twin experiment. All fields as well as $\{x, y, t\}$ were scaled by the values in the table, so all calculations were done with dimensionless variables.

model output with the data (Whartenby *et al.*, 2013). In other words, the height field and at least one of the velocity fields at each grid point needed to be observed.

Since these results were roughly consistent among the three resolutions tested, we restrict our discussion here to the case where $N_\Delta = 16$. Representative plots of the height and velocity fields are shown in Figure 1. The total number of degrees of freedom is $D = 3N_\Delta^2 = 768$, for which Whartenby *et al.* (2013) estimated $L_s \approx 524 = 0.68\,D$. Based on the discussion above and the lectures by Cardinali (2013), we see that this requirement, which is expected to be even higher in practice, exceeds the number of measurements now available by at least a factor of two.

## 5   Results with Time Delay Nudging for the Shallow Water Equations

We now demonstrate that the time delay method is capable of reducing $L_s$, by showing that it can construct successful estimates and predictions without directly observing the horizontal velocity fields. This strategy was shown to fail by Whartenby *et al.* (2013) with static ($D_M = 1$) nudging. Thus, we assume height measurements alone are made at each grid point $(i, j)$ for $i, j = \{1, 2, \dots, 16 = N_\Delta\}$, so $L = 256 < 524 \approx L_s$, as estimated by Whartenby *et al.* (2013).

The initial state $\mathbf{x}(t_0)$ for the model and the data are taken to have the form,

$$h^{(i,j)}(t_0) = \left(\frac{\pi A_0}{N_\Delta \Delta Y}\right)^2 \left[\cos(\omega_\phi \, \phi(\mathbf{r}^{(i,j)}) + \delta_\phi) + \cos(\omega_\theta \, \theta(\mathbf{r}^{(i,j)}) + \delta_\theta)\right] + H_0$$

$$u^{(i,j)}(t_0) = A_0 \frac{\partial \psi(\mathbf{r}^{(i,j)})}{\partial y} \qquad v^{(i,j)}(t_0) = -A_0 \frac{\partial \psi(\mathbf{r}^{(i,j)})}{\partial x} \tag{11}$$

where the parameters $H_0 = 5100m$, $A_0 = 10^6$ and

$$\psi(\mathbf{r}) = \cos(\omega_\phi' \phi(\mathbf{r}) + \delta_\phi') \sin(\omega_\theta' \theta(\mathbf{r}) + \delta_\theta'). \tag{12}$$

The functions $\phi$ and $\theta$ respectively evaluate the latitude and longitude at the point $\mathbf{r}^{(i,j)}$ on the grid. All fields as well as the variables $\{x, y, t\}$ were scaled by the values in Table (1), to make them dimensionless. The parameters $\omega_\phi, \omega_\theta, \omega_\phi', \omega_\theta'$ and $\delta_\phi, \delta_\theta, \delta_\phi', \delta_\theta'$ are chosen arbitrarily, so that the phase and period of the initial condition are different for truth and the estimate. Also, although the method is capable of estimating the static model parameters, here they are considered known.

The coupling matrix $\mathbf{G}(t)$ is taken to be diagonal, with different weights for the heights and for the velocities. In particular, $G_{u,v} \Delta t = 0.5$ and $G_h \Delta t = 1.5$ with $\Delta t = 0.01 h = 36 s$. The values of $G_h$ are larger than $G_u, G_v$, since the average height is $5000 \pm 30 m$, three orders of magnitude higher than the average velocity $0 \pm 5 m/s$. The time delay space coupling $\mathbf{g}(t)$ is taken to be the identity matrix, as all the height measurements are assumed to be known with equal temporal precision throughout the observation window.

The time delay was selected to be $\tau = 10 \Delta t = 0.1 h$, in order to maintain a balance between numerical stability and the common criterion of independence between the components of $\mathbf{S}(\mathbf{x}(t))$. The first minimum of the average mutual information was also calculated to be $\tau \approx 30 \Delta t$ using the method given by Abarbanel (1996). This is reasonably close to our choice, and the results did not change if its value was shifted by a few $\Delta t$.

## 5.1 Choosing $D_M$

The state was estimated by integrating the coupled differential equations Eq. (8) from $t = 0$ to $T = 5 h = 500 \Delta t$ with various $D_M = \{1, 6, 8, 10\}$. The coupling terms were then switched off at $t = T$ to generate predictions until $t = 500 h$.

Short and long term synchronization error Eq. (9) trajectories $SE(t)$ are plotted in Figure 2 for various $D_M$. Choosing $D_M = \{1, 6\}$, yields a synchronization error that remains around its initial value of $0.005$ until the end of the five hour observation window. After the coupling is switched off, the error rises very rapidly until stabilizing around $0.1$ for the remainder of the prediction window. By contrast, for $D_M = \{8, 10\}$ the synchronization error falls steeply to order $10^{-6}$ within the observation window. It then subsequently rises as $\exp[\lambda_{max}(t - T)]$, where $\lambda_{max} \approx 1/31 h$ agrees with the largest Lyapunov exponent calculated for this flow. This exponential rate of growth is particularly evident in the long trajectory displayed in the **Right Panel** of Figure 2.

Since $D_M \geq 8$ produces error values several orders of magnitude smaller than those obtained with $D_M \leq 6$, we expect the state estimates $\mathbf{x}(T)$ obtained with $D_M \geq 8$ to be quite accurate when compared with the estimates for $D_M \leq 6$. These estimates are now evaluated as they would be in a true experiment, by comparing predictions on the observed heights with additional data. In Figure 3 the known (black), estimated (red), and predicted (blue) height trajectories are shown for an arbitrarily selected grid point $h^{(6,4)}(t)$. Short and long term prediction trajectories computed with $D_M = 6$ are displayed in Figure 3 upper panels respectively. Corresponding results for $D_M = 8$ are shown in the lower panels. As anticipated, the predictions for $D_M = 8$ are clearly superior to those obtained with $D_M = 6$. In addition, with the choice $D_M = 7$ some initial conditions synchronized while others did not. Further analysis of this 'boundary' case is an interesting area for future study.

The failure of predictions obtained with $D_M = 6$ is a result of poor estimates of the unobserved states (i.e. fluid velocities) at $t = T$. Although in an actual experiment we would not be able to verify this statement directly, we may do so here. Velocity profiles $u^{(6,4)}(t)$ displaying short and long time comparisons between the known (black), estimated (red) and predicted (blue) values are given in Figure 4 for $D_M = 6$ in the upper panels, and for $D_M = 8$ in the lower panels. We find the situation is indeed as anticipated; the estimates and predictions are quite unacceptable for $D_M = 6$, whereas for $D_M = 8$ they are highly accurate. The same improvement in predictive accuracy was obtained for the other velocity component $v^{(6,4)}(t)$. These results

are plotted in Figure 5.

Predictions were also calculated for $D_M = 1$ and $D_M = 10$, but these results are not shown. They agree with the synchronization error calculations in Figure 2, in that the predictions generated with $D_M = 10$ are just as accurate as those for $D_M = 8$. Likewise, predictions with $D_M = 1$ (i.e. simple nudging) are very poor, in accordance with Whartenby *et al.* (2013).

## 5.2    Reducing the coupling strength

In the previous discussion it was suggested that reducing the coupling strength will have a detrimental effect on the quality of the estimation procedure and the resulting prediction. We investigate this now, by performing the same calculations as above with $D_M = 10$ but reducing the coupling on the height so we have $G_h \Delta t = G_u \Delta t = G_v \Delta t = 0.5$. The synchronization error $SE(t)$, shown in Figure 6, **Upper Left Panel**, stabilizes to a level three orders of magnitude higher than was achieved with $G_h \Delta t = 1.5$, suggesting failure. This is confirmed in the remainder of Figure 5, which displays the known (black), estimated

(red) and predicted (blue) values for $h^{(6,4)}(t)$, $u^{(6,4)}(t)$, and $v^{(6,4)}(t)$, respectively. Although the height estimate is rather good and the prediction is not terrible, at least for the first 15-20 hours after the end of the assimilation window, the unobserved states are clearly not well estimated at any time $t \leq T$.

This result demonstrates that proper choice of coupling is required. However, we have not developed a systematic way of choosing these values, and it is known from classical results on synchronization that the optimal choice depends on the number

and distribution of observations. Furthermore, the fact that the height estimates appear to be rather accurate also emphasizes the point that, in a true experiment, the success of the assimilation procedure must be evaluated against the forecasts—not the analyses.

## 5.3    Further reducing the number of measurements

In addition, until now we have conveniently chosen to observe the height field at all $L = N^2 = 256$ grid locations. We now

attempt to reduce $L$ even further, by repeating the analysis with $L = 252$ and $L = 248$ height measurements, chosen at arbitrary grid points. From the results displayed in the **Upper Left Panel** of Figure 7, it is evident that for $L = 252$ rapid and accurate synchronization is still achieved, while for $L = 248$ it is not. In addition, the known (black), estimated (red), and predicted values (blue) for $h^{(6,4)}(t)$ are shown in the other panels of Figure 7 for $L = 248$ and $L = 252$ respectively. Results for the unobserved velocity fields agree as well, though these results are not shown.

Thus, even with time delays, it may not be possible to significantly reduce the number of required height measurements. We remark however, that the overall space of parameters appearing in our study has not been thoroughly explored. Additional

refinement of the parameters $\mathbf{G}(t)$, $\mathbf{g}(t)$, $D_M$, and $\tau$ may further reduce this constraint, for instance by allowing $\mathbf{G}(t)$ to be non-diagonal.

## 5.4 Noise in the observations

We now repeat the above calculations for $L = 252$ with Gaussian noise $N(0, \sigma)$ added to the height observations. A comparison is shown in Figure 8 for $\sigma = \{0.2, 0.5\}$ and $D_M = \{8, 10\}$. The synchronization error still falls rapidly within the observation window, although not to $O(10^{-5})$, as in the noiseless case. In the prediction window, it rises in an exponential manner as expected. Furthermore, results fail to synchronize when the magnitude of the noise gets large enough. This transition occurs at roughly $\sigma = \{1.3, 2.0\}$ for $D_M = \{8, 10\}$ respectively. These results were included to show that the method appears to be relatively robust to small errors in the observations. A more thorough examination of the impact of imperfect observations will be given elsewhere.

## 5.5 Using drifter data

Another quite important source of observations about ocean flows is being provided by position measurements $\mathbf{r}(t)$ of Lagrangian drifters (Mariano *et al.*, 2002). Such observations have been shown to be a good supplement to the traditional observations made on a fixed grid (Kuznetsov *et al.*, 2003) and they can also be used to estimate an Eulerian velocity field (Molcard *et al.*, 2003; Piterbarg *et al.*, 2008; Salman *et al.*, 2006). In this section, we combine the time delay method with a data set from drifter measurements to show that they can provide accurate estimates for the grid state variables $\{h(\mathbf{r}^{(i,j)}, t), u(\mathbf{r}^{(i,j)}, t), v(\mathbf{r}^{(i,j)}, t)\}$, without much additional effort.

We monitor the positions of $N_D$ drifters deployed at randomly chosen grid locations and afterwards allowed to move freely to provide spatially continuous measurements between grid points. Drifters were also deactivated when they reached the boundary of the grid, so the number of operational drifters decreases throughout the estimation window. The dynamics of drifters are approximated as two-dimensional fluid parcel motion near the surface of the water layer, which are determined by the Lagrangian equations

$$\frac{d\mathbf{r}^{(n)}(t)}{dt} = \mathbf{u}(\mathbf{r}^{(n)}(t), t) \tag{13}$$

where $\mathbf{r}^{(n)}(t)$ is the position of the $n^{th}$ drifter and this equation was simulated by linear interpolation of the discrete velocity fields (Press *et al.*, 2015; Thompson & Emery, 2014). Hybrid measurements are incorporated into the time delay nudging method by combining the grid variables and the collective drifter positions

$$\xi^{\dagger}(t) = \{[\mathbf{r}^{(1)}(t)]^{\dagger}, [\mathbf{r}^{(2)}(t)]^{\dagger}, \ldots, [\mathbf{r}^{(N_D)}(t)]^{\dagger}\}$$

into a single hybrid state vector. The corresponding time delayed vectors are $\mathbf{Y}_{drifter}(t) = \{\mathbf{Y}_{grid}(t), \mathbf{Y}_{drifter}(t)\}$ and $\mathbf{S}_{drifter}(t) = \{\mathbf{S}_{grid}(t), \mathbf{S}_{drifter}(t)\}$ respectively, where

$$\mathbf{Y}_{drifter}^{\dagger}(t) = \{\xi_{data}^{\dagger}(t), \xi_{data}^{\dagger}(t + \tau), \ldots, \xi_{data}^{\dagger}(t + \tau(D_M - 1))\}$$

$$\mathbf{S}_{drifter}^{\dagger}(t) = \{\xi_{model}^{\dagger}(t), \xi_{model}^{\dagger}(t + \tau), \ldots, \xi_{model}^{\dagger}(t + \tau(D_M - 1))\}.$$

We consider cases with $N_D = 0$ (no drifters), as well as $N_D = 20$ and $N_D = 64$, in addition to $L = 208$ and $L = 128$ grid observations. Example plots showing the initial locations for $N_D = 20$ and $N_D = 64$ are given in Figure 9. While many runs were successful using the same setup described above, the results were somewhat dependent on where the drifters were initialized. These results are not shown.

The consistency of the results improved by choosing the initial estimate to only in magnitude from the true solution, rather than in both phase and frequency as was done above. Specifically, for the results reported below the initial conditions of the data $\psi_{data}(\mathbf{r}^{(i,j)}, t_0)$ and $h_{data}(\mathbf{r}^{(i,j)}, t_0)$ and of the model $\psi_{model}(\mathbf{r}^{(i,j)}, t_0)$ and $h_{model}(\mathbf{r}^{(i,j)}, t_0)$ are related by $\psi_{data}(\mathbf{r}^{(i,j)}, t_0) = C_0 \psi_{model}(\mathbf{r}^{(i,j)}, t_0)$ and $h_{data}(\mathbf{r}^{(i,j)}, t_0) = C_0 h_{model}(\mathbf{r}^{(i,j)}, t_0)$. We choose $C_0 = 1.0 + 0.1 \eta$, with $\eta$ selected from a uniform distribution in the interval $[-1, 1]$. The velocity fields are found as above, using $\psi(\mathbf{r}, t_0)$ as a stream function.

In Figure 10, we show the synchronization error of observed quantities when for $D_M = 8$, keeping all other parameters the same as in the previous calculations. We present (in red) the synchronization error for L = 208 height observations and $N_D = 20$ drifter observations, and we show (in blue) the same synchronization error when $L = 208$ and $N_D = 0$ drifters are deployed. With $L = 208$, namely, observing 27% of the heights and 20 drifters, the synchronization error converges to a small value within the five hour observation window. Without drifters, the estimation fails. Furthermore, by increasing the

number of drifters to $N_D = 64$ within a 30 minute observation window, synchronization can be achieved with $L = 128$ height observations. Snapshots of the fields at different times throughout the estimation and prediction window are shown in Figure 11 for comparison.

Although we have not yet explored how to balance the number of drifters and the number of height (or other) measurements, these preliminary results suggest positional data from drifters can be useful for improving the observability of the system.

In contrast to other approaches in which the drifter data is used to directly interpolate the grid variables (Kuznetsov *et al.*, 2003), our method transfers positional information from the drifters to the estimate through the dynamical model. Whether this approach is valid for real data remains to be seen however. Furthermore, time delay method provides a natural way to incorporate this information into the analysis.

## 6  Discussion and Summary

The transfer of information from measurements of a chaotic dynamical system to a quantitative model of the system is impeded when the number of measurements at each measurement time is below an approximate threshold $L_s$, which can be established in a twin experiment. Whartenby *et al.* (2013) previously showed that for a nonlinear model for shallow water flow, a standard nudging technique given by Eq. (2) requires direct observation of roughly 70% of the dynamical variables $\{h(\mathbf{r}, t), u(\mathbf{r}, t), v(\mathbf{r}, t)\}$ at each measurement time to synchronize the model output with the observations.

Here we have demonstrated how information in the time delays of the observations may be used to reduce this requirement to about 30%, in which only the height fields need be observed. Moreover, it appears $L_s$ can be even further reduced by adding positional information from drifters, which interpolate the height field at locations between grid points.

Although all this has been done on a simplified model of shallow water flow, implemented with only $D = 3N_\Delta^2 = 768$ degrees of freedom, the process can be used to analyze increasingly realistic and complex models of coupled earth systems. Since the successful analysis of simulated data is a prerequisite for success with real data, this methodology provides a way to assess where one stands with respect to critical observability limits of the system at hand.

Furthermore, we expect that this formalism will generalize to systems substantially larger than the one presented here, although we do not underestimate the numerical challenges involved in its extension to say, the scale of operational NWP models. We also suspect this issue of insufficient measurements to be a critical limitation in our current ability to predict the behavior of complex, chaotic systems. Since such systems are quite typical in practice, these issues need to be examined with more realistic models.

Harking back to the introduction, we note that in the report by Cardinali (2013) indicates that 30-40 million daily observations are now available at the ECMWF, and that many NWP models comprise upwards of $10^8$ degrees of freedom. If the qualitative trends shown here, in which time delays provide successful predictions with only 30% of the state variables observed, can be extended to substantially larger systems, then this method may indeed be useful for improving the forecasts of existing operational NWP models.

*Acknowledgements.* This work was funded in part under a grant from the US National Science Foundation (PHY-0961153). Partial support from the Department of Energy CSGF program (DE-FG02-97ER25308) for D. Rey is appreciated. Partial support from the MURI Program (N00014-13-1-0205) sponsored by the Office of Naval Research is also acknowledged. We would also like to thank the reviewers for their thorough reading and thoughtful suggestions that have helped significantly improve this manuscript.

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

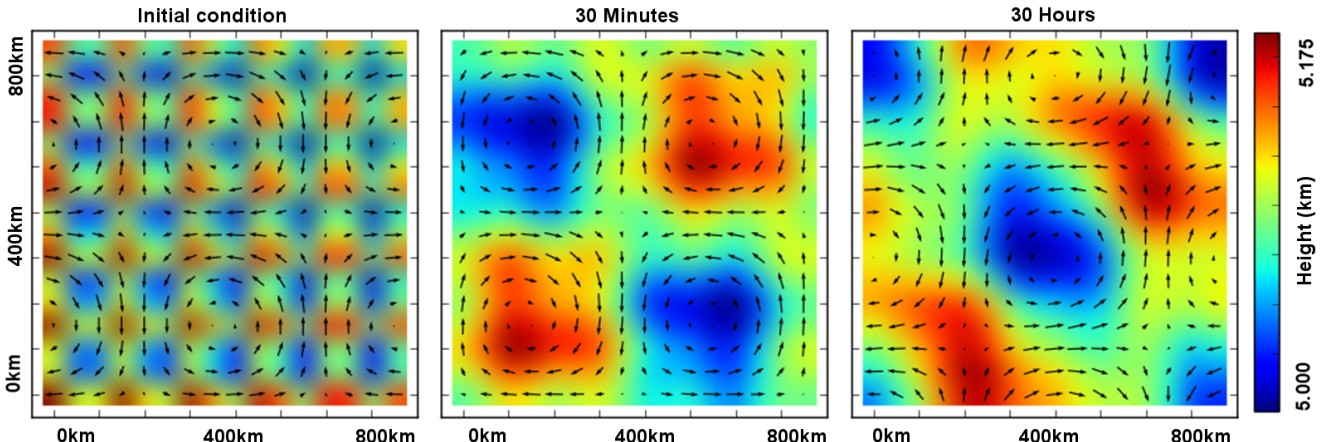

**Figure 1.** Snapshots of shallow water flow on a $16 \times 16$ grid spanning $400$ km on each side. Heights and velocity fields are shown: **Left Panel** at the initial time, **Center Panel** after 30 minutes, and **Right Panel** after 30 hours.

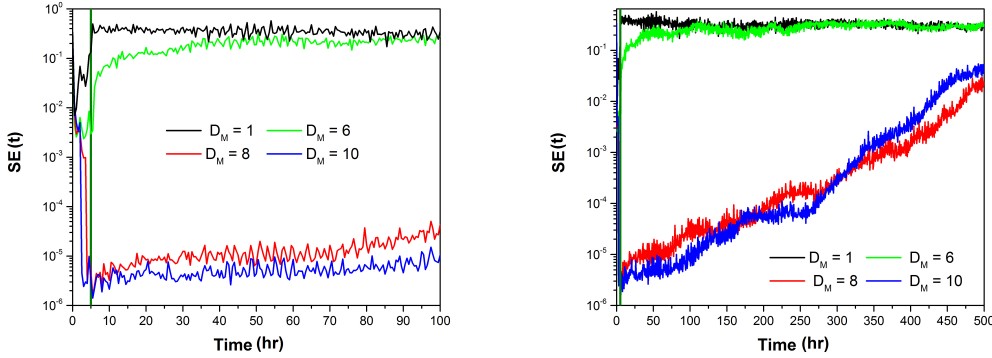

**Figure 2.** Synchronization error $SE(t)$, defined in Eq. (9), computed with $D_M = \{1, 6, 8, 10\}$, $G_h \Delta t = 1.5$, $G_u \Delta t = g_v \Delta t = 0.5$ and $\tau = 10 \Delta t = 0.1 h$. Assimilation is performed for $t \leq 5$ hr. **Left Panel** The couplings are then switched off and predictions are generated using the original dynamical equations Eq. (10) until $t = 100 h$. In the prediction window ($t \geq 5$), the error in the trajectories grow roughly with the largest Lyapunov exponent of the system $\lambda_{max} \approx 1/31 h$. Synchronization is evident when $D_M = \{8, 10\}$ and not for $D_M = \{1, 6\}$, suggesting that accurate predictions will be obtained $D_M = \{8, 10\}$. **Right Panel** The same calculation, but extended to t = 500 hr.

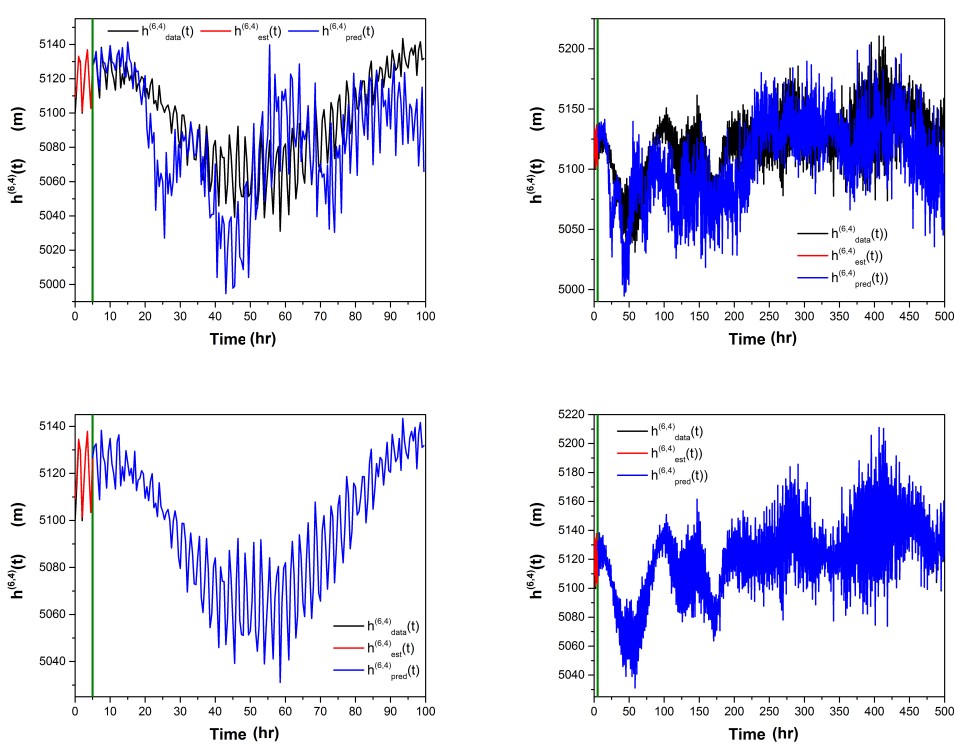

**Figure 3. Upper Left Panel** Known (black), estimated (red) and predicted (blue) for the observed height values $h^{(6,4)}(t)$ at grid point $(6,4)$ for $D_M = 6$. Observations are for $0 \leq t \leq 5$ hr. Predictions are for $5 \leq t \leq 100$ hr. **Upper Right Panel** The same calculation for $D_M = 6$ for a longer prediction window $5 \leq t \leq 500$ hr. **Lower Left Panel** The same calculation except $D_M = 8$. Prediction window is $5 \leq t \leq 100$ hr. **Lower Right Panel** The same calculation except $D_M = 8$. Prediction window is $5 \leq t \leq 500$ hr.

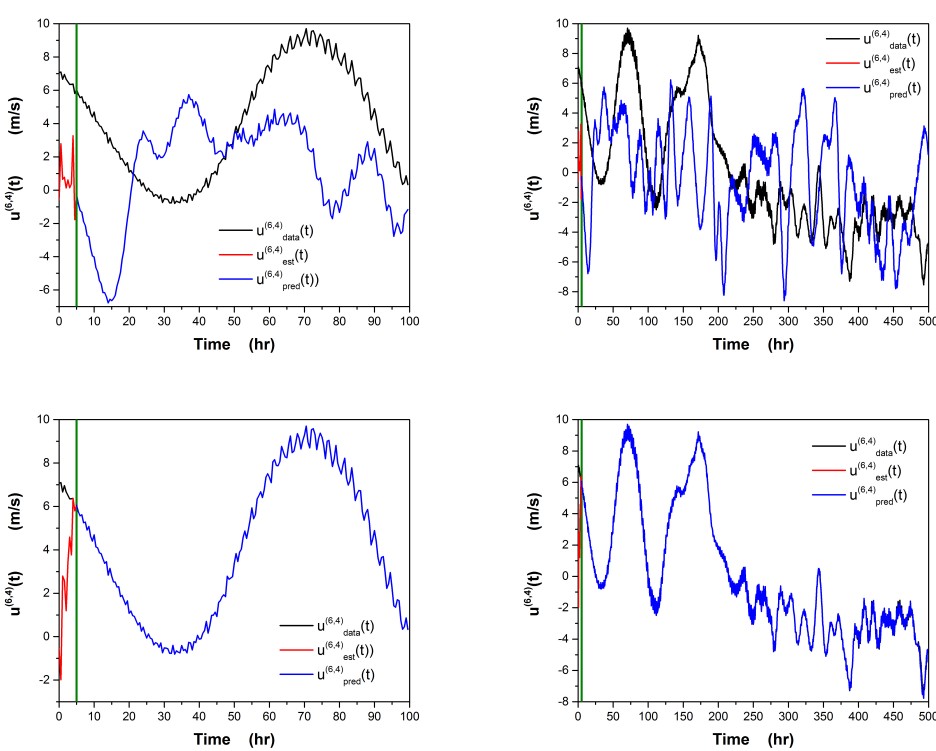

**Figure 4. Upper Left Panel** Known (black), estimated (red) and predicted (blue) for the observed x-velocity values $u^{(6,4)}(t)$ at grid point $(6,4)$ for $D_M = 6$. Observations are for $0 \leq t \leq 5$ hr. Predictions are for $5 \leq t \leq 100$ hr. **Upper Right Panel** The same calculation for $D_M = 6$ for a longer prediction window $5 \leq t \leq 500$ hr. **Lower Left Panel** The same calculation except $D_M = 8$. Prediction window is $5 \leq t \leq 100$ hr. **Lower Right Panel** The same calculation except $D_M = 8$. Prediction window is $5 \leq t \leq 500$ hr.

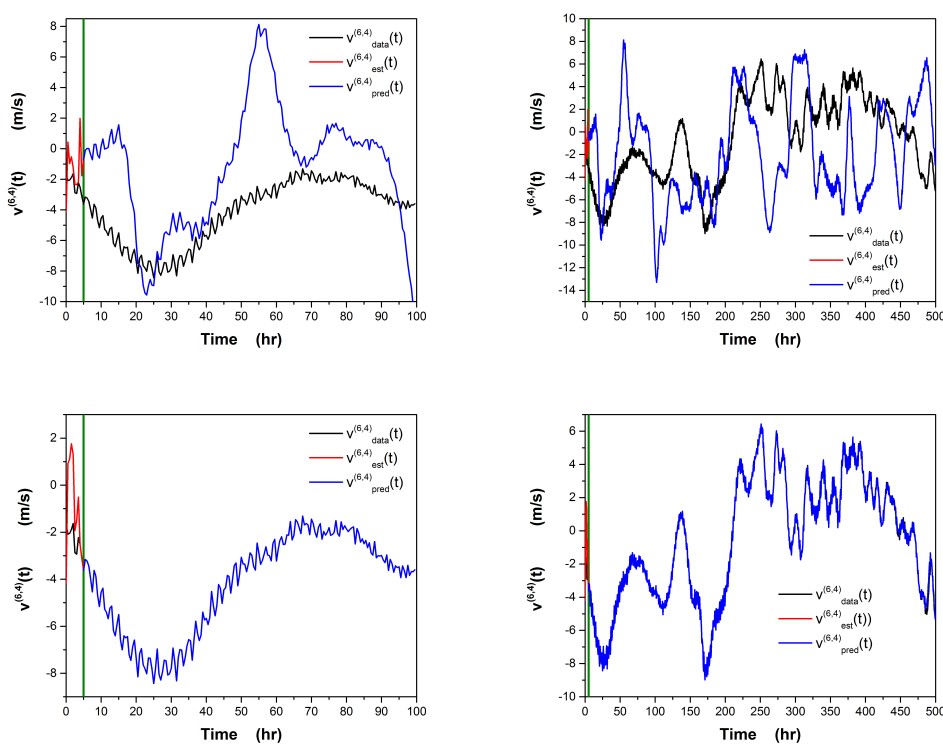

**Figure 5. Upper Left Panel** Known (black), estimated (red) and predicted (blue) for the observed y-velocity values $v^{(6,4)}(t)$ at grid point $(6,4)$ for $D_M = 6$. Observations are for $0 \leq t \leq 5$ hr. Predictions are for $5 \leq t \leq 100$ hr. **Upper Right Panel** The same calculation for $D_M = 6$ for a longer prediction window $5 \leq t \leq 500$ hr. **Lower Left Panel** The same calculation except $D_M = 8$. Prediction window is $5 \leq t \leq 100$ hr. **Lower Right Panel** The same calculation except $D_M = 8$. Prediction window is $5 \leq t \leq 500$ hr.

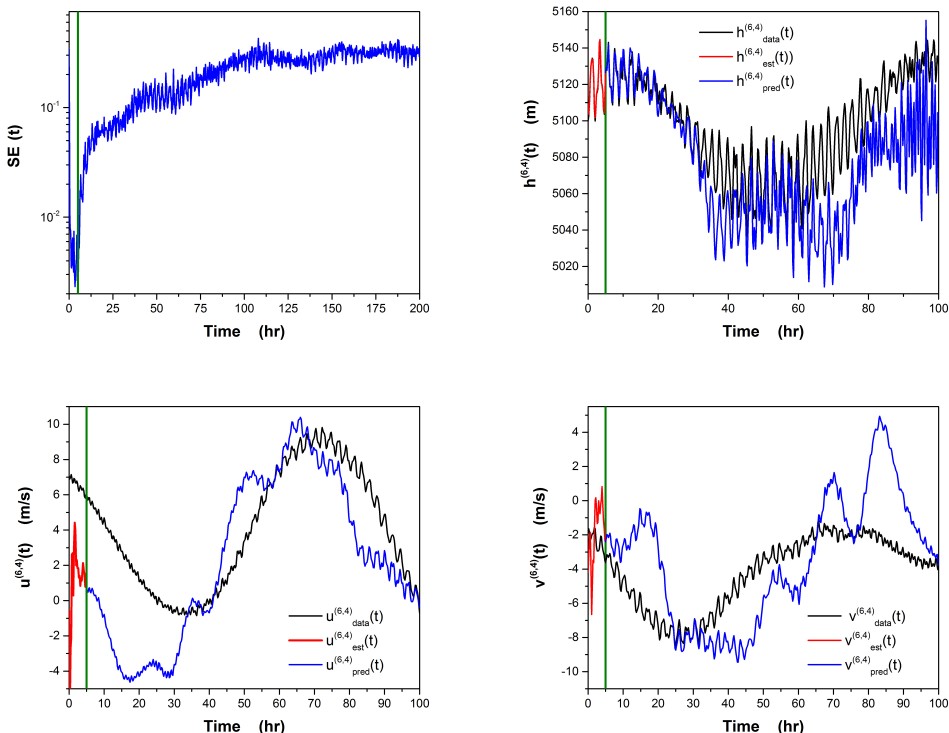

**Figure 6.** Data assimilation results with $D_M = 10$ and reduced coupling on the height component $h^{(6,4)}(t)$ at location (6,4), $g_h \, \Delta t = g_u \, \Delta t = g_v \, \Delta t = 0.5$. All other parameters are the same. **Upper Left Panel** $SE(t)$ for $0 \le t \le 200$ hr. **Upper Right Panel** Known (black), estimated (red) and predicted (blue) for the observed height values $h^{(6,4)}(t)$ at grid point $(6,4)$ for $D_M = 10$. Observations are for $0 \le t \le 5$ hr. Predictions are for $5 \le t \le 100$ hr. **Lower Left Panel** Known (black), estimated (red) and predicted (blue) for the observed x-velocity values $u^{(6,4)}(t)$ at grid point $(6,4)$ for $D_M = 6$. Observations are for $0 \le t \le 5$ hr. Predictions are for $5 \le t \le 100$ hr. **Lower Right Panel** Known (black), estimated (red) and predicted (blue) for the observed y-velocity values $v^{(6,4)}(t)$ at grid point $(6,4)$ for $D_M = 6$. Observations are for $0 \le t \le 5$ hr. Predictions are for $5 \le t \le 100$ hr.

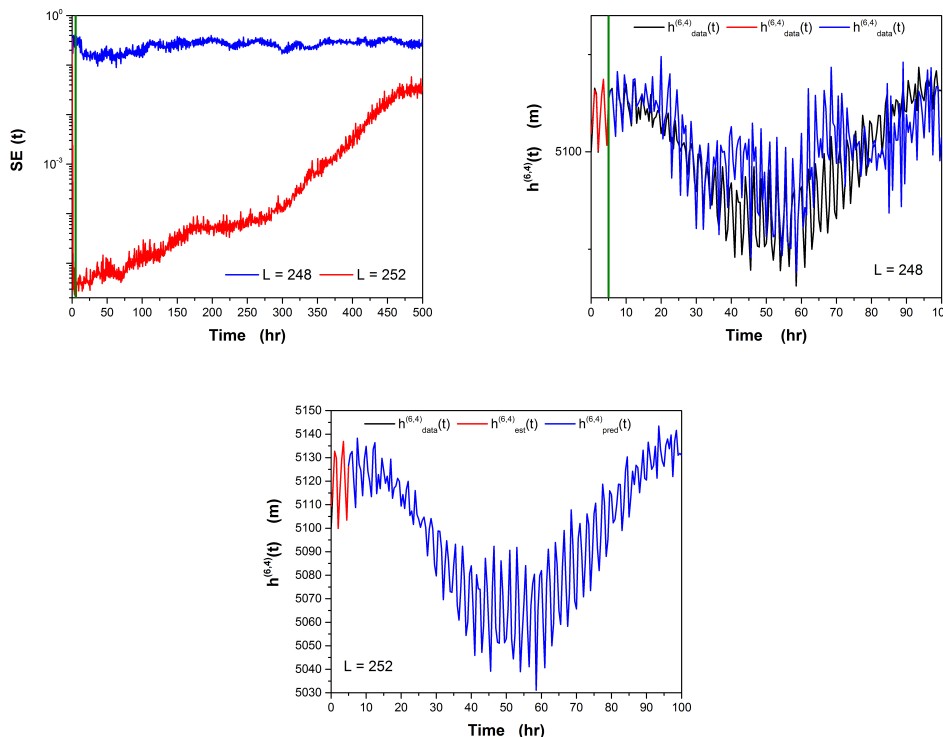

**Figure 7.** Synchronization error and known, estimated, and predicted height values for L = 248 height measurements at each observation time and for L = 252 height measurements at each observation time. **Upper Left Panel** $SE(t)$ for L = 248 and L = 252 over $0 \leq t \leq 5$ h in the observation window, and $5 \leq t \leq 500$ h after the couplings are removed. **Upper Right Panel** Known (black), estimated (red), and predicted (blue) values of the height $h^{(6,4)}(t)$ at gridpoint (6,4) for $0 \leq t \leq 100$ h for L = 248. **Lower Panel** Known (black), estimated (red), and predicted (blue) values of the height $h^{(6,4)}(t)$ at gridpoint (6,4) for $0 \leq t \leq 100$ h for L = 252. This shows the rather sharp transition between bad predictions (L = 248) and good predictions (L = 252).

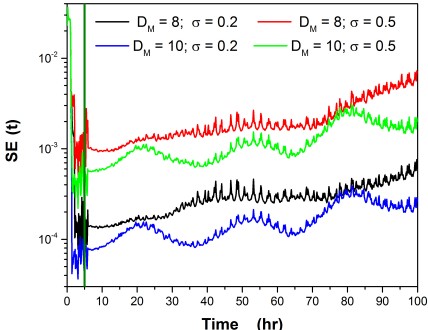

**Figure 8.** The effect of noise levels in the initial condition for the solution of the model equations Eq. (10) on $SE(t)$. We show the results for $D_M = 8$ and 10 for added Gaussian noise $N(0, \sigma)$ with $\sigma = 0.2$ and 0.5. For this range of noise levels added to the initial condition for generating the data in our twin experiments, we see that the detailed values of $SE(t)$ change. In the case of both $D_M = 8$ and $D_M = 10$, $SE(t)$ still becomes quite small in the observation window $0 \leq t \leq 5$ h, suggesting that predictions for $t \geq 5$ will remain robustly accurate.

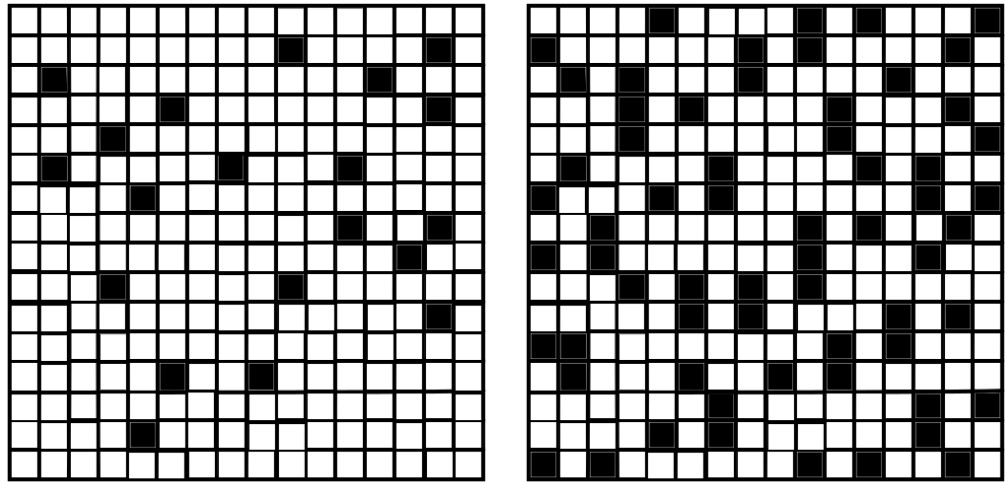

**Figure 9.** Initial positions for **Left Panel** $N_D = 20$ drifters and **Right Panel** $N_D = 64$ drifters.

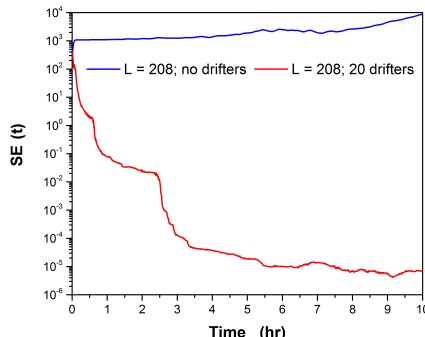

**Figure 10.** $SE(t)$ for our standard twin experiment described in detail earlier when we utilize drifter information, and when we do not utilize drifter information. When the number of observations of height is L = 208, we see that without drifter information (blue line) there is no synchronization and correspondingly inaccurate predictions (not shown). When information from 20 Lagrangian drifters is added during data assimilation using time delay nudging, $SE(t)$ decreases very rapidly (red line) indicating predictions will be very accurate (also not shown). The efficacy of small numbers of drifters is clear in this example.

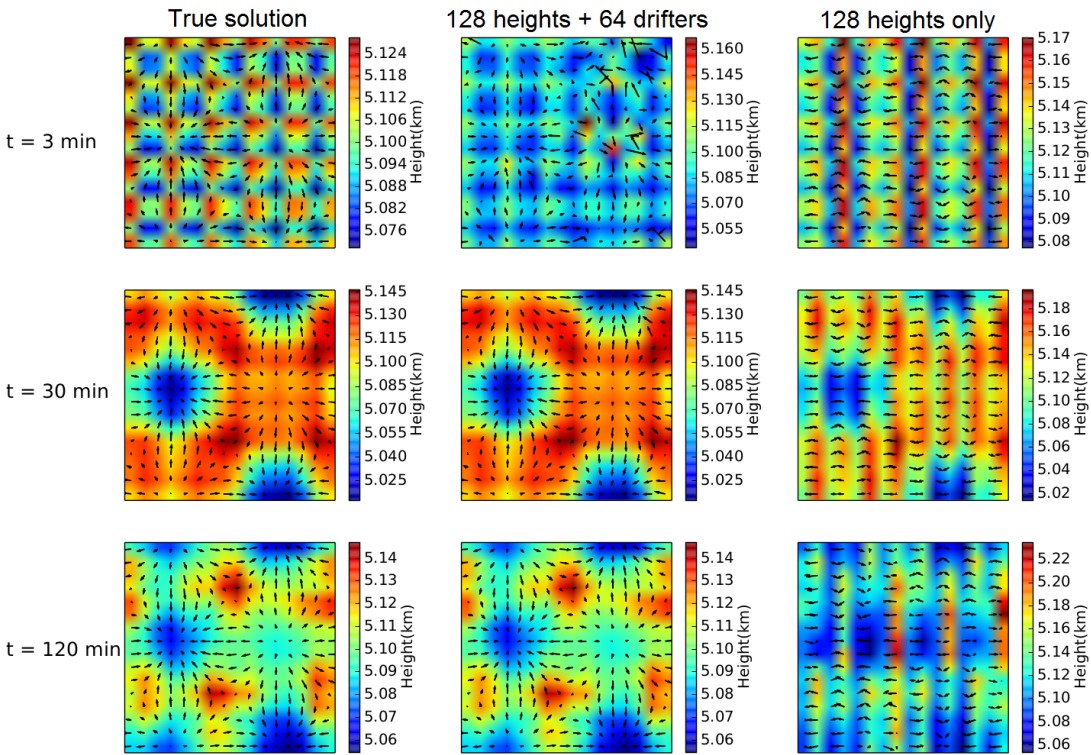

**Figure 11.** Comparison of the estimated and predicted fields $\{\mathbf{h}(t), \mathbf{u}(t), \mathbf{v}(t)\}$ between the truth (**Left Column**) and analyses, run with observations of 128 height variables, both with (**Center Column**) and without drifters (**Right Column**). Snapshots are taken 3 min (**Upper Row**) into the assimilation window, at 30 min the end of the assimilation window (**Center Row**), and 90 min into the prediction window (**Bottom Row**).