# Peer review of "Estimating the State of a Geophysical System with Sparse Observations : Time Delay Methods to Achieve Accurate Initial States for Prediction"

_Nonlinear Processes in Geophysics, 2016_

## Referee Comment (RC1) · S.G. Penny (Referee) · 2 May 2016

Overview evaluation:

Techniques for dealing with a sparse observational networks are critically important, particularly for ocean and climate reanalyses that attempt to reconstruct the past state of the Earth system (e.g. Compo et al., 2011; http://www.esrl.noaa.gov/psd/data/gridded/data.20thC_ReanV2.html). The experiment scenarios described here by the authors are perhaps most applicable to the estimation of the global ocean state after the introduction of satellite altimeters, e.g. TOPEX/Poseidon in late 1992 (https://sealevel.jpl.nasa.gov/missions/topex/), with their final set of experiments having a potential application to leverage data from the Global

CC-BY license logo

Drifter Program (http://www.aoml.noaa.gov/phod/dac/index.php). Thus from a practical point of view, the time-delay method has potential merit for operational scale data assimilation (DA) and reanalysis.

Because of such potential, the authors should give a bit more explicit description about how these ideas compare to common methods like 4DVar or the 4D Ensemble Kalman Filter (EnKF), both of which utilize observations over an extended time window. The authors could give a more thorough depiction of how their ideas could be incorporated in these existing systems in order to facilitate a higher likelihood that an operational center might adopt the approach.

The sea surface height is closely connected to the near surface currents via the geostrophic balance, particularly in midlatitudes. Thus it is expected that unobserved currents would be well constrained by proper estimation of the surface height. For example, sea surface heights and sea surface winds are used to construct an estimate of ocean surface currents for the OSCAR product (http://www.oscar.noaa.gov/index.html). However, the examples given by the authors could perhaps be described as a supplement for the tropical region where this relationship breaks down. For future work, a natural extension would be to address a slightly more sophisticated example consisting of multiple vertical layers and the modeling of the temperature and salinity components of the density. This experiment would give a better test of estimating unobserved variables. For example, observing only temperature while estimating salinity is a challenging problem for ocean reanalyses before the Argo era.

A brief statement could be made about the applicability of the time-delay approach, for example, to the tropical observing system of moored buoys (TAO/TRITON). These are stationary sensors generating data about once every 10 minutes, but the majority of this data is not used in DA because most global scale ocean assimilation systems use analysis cycles that span multiple days. Even a coupled ocean/atmosphere DA system cycling every 6 hours could benefit from better use of this data. I suggest investigating the TPOS-2020 (Tropical Pacific Observing System) effort for the potential to inform

the future development of this and other observing systems (http://tpos2020.org).

A weakness in the chosen experiments scenarios that should be acknowledged is that the approach has not been tested on time-delay observations with errors that are correlated in the time dimension. This is particularly important in ocean DA because errors of representativeness often dominate (versus instrument errors). I suggest an experiment, perhaps for future work, in which you run 2 model resolutions. The high resolution run is treated as 'truth', from which observations are drawn. The low resolution model is what you are synchronizing via DA. Set up appropriately, this should give you 'natural' errors of representativeness in the observations that may be correlated in time with the errors of future or past observations. Does the time-delay method still work effectively in this experiment scenario?

The time-delay method is described in comparison to nudging as a baseline. I would like to see the authors compare a simple 4DVar to the time-delay method as well (via experiment) to give context into how their method compares to a more state-of-the-art DA. It seems that the time-delay information for the observations and model state applied with what is essentially a diagonal coupling term emulates a similar effect as the cross-covariances that would in effect apply a non-diagonal coupling term to the innovations computed at different times throughout the window. The authors should discuss how the off-diagonal coupling used in most operational DA relates to the diagonal coupling with time-delay observations used here.

The impact of observation error on synchronization via the nudging approach is not addressed very thoroughly. I'd like to see some evaluation of the sensitivity to observation error in the assessment of the method. The authors should describe how their method is impacted by outliers in the observed data. Is the method sensitive to such outliers? I'd like to see an example.

General Technical Corrections:

The manuscript would benefit from more effort in bridging the nonlinear dynamics synchronization terminology with standard data assimilation terminology. This would particularly benefit the readership of this journal.

There is an odd mixing of tenses throughout the citations, and the inconsistency is distracting. The focus should be on the work presented here. Example substitutions are given in the specific comments below. Also, if the author name in the citation is not used as a part of the sentence, please put it in parentheses.

The discussion of 4DVar from the conclusion section would be much more useful either in the Introduction or in its own section right after the introduction.

The conclusion section is disorganized and needs attention. It should not introduce new concepts in any great detail - instead those should be moved to the body of the paper. Page 14, Lines 24 through the end seem to be a good start to the conclusion section, I advise starting there and reorganizing the rest.

The figures are insufficient. I would like to see a graphical example of the 2-Dimensional domain used for the experiments, including the spatial scale of the layer heights and velocity field. For the drifter experiments, the initial locations for the various experiments should be plotted. Overall, better use of visuals/graphics would greatly enhance the communication.

Specific Technical Corrections:

Page 1:

Abstract:

Line 1:

"The data assimilation process, in which observational data is used to estimate the states and parameters of a dynamical model,"

This is a narrow definition of modern data assimilation. Perhaps reword to indicate that this is one application of data assimilation.

Line 4:

"Since this problem of insufficient measurements is typical across many fields, including numerical weather prediction,"

It's not clear in what manner the authors feel the measurements are insufficient. A large number of atmospheric measurements are often discarded in operational DA. Is it the quantity or type of observations in the atmosphere that is lacking? The sparsity of observations is certainly a challenge in ocean DA, but this must be considered relative to the timescales and spatial scales of interest. Could the authors please clarify.

Line 5:

"introduced in Rey et al (2014a, b)"

Change to: "introduced [by] Rey et al."

Line 8:

"For instance, in Whartenby et al (2013) we found that to achieve this goal, standard nudging requires observing approximately 70% of the full set of state variables. Using time delays, this number can be reduced to about 33%, and even further if Lagrangian drifter information is also incorporated."

Change to:

"While Whartenby et al (2013) we found that standard nudging requires observing approximately 70% of the full set of state variables, we find that this number can be reduced to about 33% using time delays, and even further if Lagrangian drifter information is also incorporated."

While I understand the comparison to nudging as a baseline/control methodology, could you give an idea of what proportion of state variables must be observed for a more sophisticated DA scheme like 4DVar or an EnKF?

Introduction:

Line 12:

"The ability to forecast the complex behavior of the earth's coupled ocean, atmosphere system lies at the core of modern numerical weather prediction (NWP) efforts."

This is true, but the current drive amongst many centers is modeling and forecasting the entire integrated Earth System, including for example atmosphere, ocean, sea-ice, land, aerosol, surface waves, ionosphere, etc. I would suggest either saying "the coupled earth system", or further specifying examples beyond simply atmosphere/ocean coupling which has been conducted at some centers for many years.

Line 14:

"the observations are completed"

I'm not quite sure what it means to 'complete' an observation. Perhaps this could be reworded.

Line 15:

"The latter is indispensable, even if one has a perfect model, as the accuracy of the prediction is crucially determined by the quality of the estimated initial state values: if the state of the model at the end of observation window is inaccurate, the forecasts will be undependable."

It is my preference to reduce the usage of emphatic adjectives in scientific writings. A possible alternative:

"Even with a perfect model, the accuracy of the prediction is dependent on the quality of the estimated initial state values. Such sensitive dependence on initial conditions was identified by Lorenz (1963; JAS)."

Line 20:

"In our earlier work Whartenby et al (2013) we showed that"

Change to:

"Whartenby et al. (2013) showed that"

Line 23:

I don't think you've specified yet what the 3 dynamical variables are. Even though it is implied by the use of the nonlinear shallow water model, I think you should be explicit.

Page 2:

Line 2:

When the authors mention the use of drifters, does that mean that they are using the position information of the drifters, or simply surface measurements of the model state? Please clarify here.

Line 3:

It is not specified whether the spatial distribution of the observations impacts the estimate of L_s. How does this estimate change when all observations are concentrated in one area, or if the observations are arranging in vertical stripes (e.g. like the TAO/TRITON array), or if they are evenly distributed (which can create aliasing artifacts in the analysis) versus randomly distributed.

There is also ambiguity about whether the state variables are reduced in number only, or in type as well. For example, a collection of global Argo profiles could be cut in half by either (a) reducing the number of floats by half, or (b) eliminating all salinity measurements and keeping only temperature. Both reduce the obs by 50%, but will have drastically different impacts. I'd ask the authors to please be more specific in how the observations are reduced.

Line 6:

"those at ECMWF Cardinali (2013)"

Change to:

"those at ECMWF (Cardinali, 2013)"

Line 7:

"Evensen (2008); Bennett (1992) should be expected"

Change to:

"(Bennett, 1992; Evensen, 2008) should be expected"

Why is Evensen 2008 ("Data Assimilation, The Ensemble Kalman Filter") cited for 4DVar? I'm sure more appropriate references could be used.

Some references that come to mind, among many others are:

Lewis and Derber, 1985: The use of adjoint equations to solve a variational adjustment problem with advective constraint.

Coutier and Talagrand, 1990: Variational assimilation of meteorological observations with the direct and adjoint shallow water equations.

Zupanski, 1997: A general weak constraint applicable to operational 4DVar data assimilation systems.

Line 7:

"While we do not discuss it here in detail, the same results for a 4DVar assimilation method Evensen (2008); Bennett (1992) should be expected."

To which same results are you referring? It is not clear. Do you expect to get all of the same results with 4DVar that you get with nudging? If so, that assertion requires some justification.

Line 11:

"The results appear to be equivalent."

To whom? This statement is ambiguous. Please state clearly whether you found this to be the case in your own research, and if this is part of your research findings, briefly present those results. Otherwise, please list citations of research showing this to be the case. Or, simply remove the statement.

Line 13:

"substantial improvement in reducing the number of observations at each measurement time that we report here."

I think you mean "reducing the number of observations needed to attain comparable accuracy in the forecasts or analysis." Please reword.

Page 3:

Line 1:

Please be careful to acknowledge that this is one application of data assimilation. Model error has many sources. Model parameter estimation can only address a small subset of the many sources of model error.

"In data assimilation we seek to use the information in observations to determine properties of a model that describes the dynamics producing those observations. These properties include unknown parameters as well as the time dependence of unobserved state variables. The model acts as a nonlinear filter coupling the observed states to the parameters and unobserved states."

Perhaps change to:

"We seek to use observations to inform a model of the dynamics producing those observations. Model parameters can be estimated along with the time dependence of unobserved state variables. The data assimilation acts as a nonlinear filter coupling the observed states to the parameters and unobserved states."

Lines 8 and 24:

Equations (1) and (2) on would be more clearly and compactly defined as matrix equations. Particularly in equation (2), the summation notation and the indices are unnecessarily busy.

Line 9:

"If the dynamics of the system is described by partial differential equations (such as with the fluids in an earth systems model) the ordinary differential equations may be realized by discretizing the partial differential equations on a grid,"

It is worth noting there is a non-trivial amount of discretization error introduced in this process.

Line 18:

"This is crucial, because it establishes a necessary condition on L that is required to synchronize the model output with the data and thereby obtain accurate estimates for the unobserved states of the system, which are also required to make good predictions."

Change to:

"This establishes a necessary condition on L that is required to synchronize the model output with the data and thereby obtain accurate estimates for the unobserved states of the system."

Line 27:

"With enough observations L, a sufficiently strong coupling will alter the Jacobian of the dynamical system Eq. (2) so that all its (conditional) Lyapunov exponents are negative."

Please write out the equations describing the Jacobian of the coupled system (2) and the requirements on its eigenvalues to lead to synchronization. I find this is not well

documented in the synchronization literature, even in the references provided. The clearest and most detailed description I have found is in the source code written by P. Bryant (http://biocircuits.ucsd.edu/pbryant/#ScientificSoftware). It would be a great contribution of this manuscript to provide a clearly documented reference for this process, particularly for the audience of this journal.

Line 29:

"This is important, as it establishes a necessary condition on L required to synchronize the model output with the data and thereby obtain an accurate estimate for the unobserved states of the system which are also needed to make good predictions."

It would be preferable if you could state specifically what necessary condition on L is required to synchronize the model. Also, this condition is dependent on the degree of nonlinearity of the system on the timescales of your observation window. A fairly linear dynamical regime would require fewer observations than a more nonlinear regime, measured perhaps using the error doubling time. More detailed discussion would be appreciated.

Line 31:

Change:

"has been shown Abarbanel et al (2009)"

to:

"has been shown [by] Abarbanel et al. (2009)"

Page 4:

Line 3:

"One way to proceed Rey et al (2014a, b); Pazo (2015) involves the recognition that additional information resides in the temporal derivatives of the observations."

This is an interesting idea, that perhaps could be most useful in static observing systems that have high temporal resolution relative to the analysis update cycle. For example, the TAO/TRITON array monitors temperatures at depth at regular 10-minute intervals. However, in NCEP's present operational ocean DA, the data are aggregated to daily or multi-day averages.

Line 9:

Since the use of waveform vs. observation time-derivative seems to be the key idea of the time-delay method in this paper, it would be nice if this mentioned up in the abstract and the introduction.

Line 10:

Please define D_M before you use it. Why is the subscript M used? Perhaps something more connected to 'time delays' could be used. Even using 'K' might make more sense since you use an index k for the time delay while you use the index l for the observations of dimension L.

Line 12:

I think it should be noted that this requires a static observing system producing observations at all points at regular intervals. How difficult would it be to extend this approach to the case when the observing system itself is dynamic?

How does tau relate to delta_t on page 1, line 19?

Line 22:

"DM need only be large enough to effectively increase the amount of information transferred from the L measurements to a value above the critical threshold, Ls."

I understand what is being said here, but it would be nice if the concept of 'amount of information' could be defined more rigorously.

Also, could the authors indicate whether they anticipate D_M to be within the analysis cycle, or perhaps longer than the analysis cycle to the point that observations are 'reused' in consecutive cycles? For example, from page 3, line 6, are the t0,t1,...tN times within the observation window the only candidates for D_M?

Line 24:

The terms "extended space measurement vector" (line 11) and "time delay model vector" (line 24) seem to refer to the same construct in two different spaces. So it would be helpful to the reader to acknowledge this similarity and use a similar terminology for both.

Line 25:

The notation for the quantity S_l,k looks like a matrix. If the authors would like it to represent a vector, it should be explained how the elements of the vector are arranged.

Lines 26-30:

You should make the bold notation consistent between here and the equations in (3) and (4). It seems perhaps you didn't mean for (3) and (4) to be bold if the indices indicate a specific element of the matrix.

Lines 25-31:

I'd like to see these equations represented in their matrix/vector form.

Line 31: Please explain the step between equations (6) and (7) in more detail.

Page 5:

Line 1:

"where repeated indices are summed over."

Which repeated indices? Just use the standard summation notation to keep it clear.

Line 1:

I think you want t to be italicized instead of boldface in G(t).

Line 23:

"So this framework tests the estimation procedure, not the model."

Change to:

"Therefore, this framework tests the estimation procedure absent of model error."

Line 24:

"Removing the issue of model error allows us to assess the weaknesses and strengths of the estimation algorithm and explore in detail the manner in which the unobserved variables are determined."

Change to:

"Removing the issue of model error allows us to assess the manner in which the unobserved variables are determined."

I would argue that the strength of the estimation algorithm cannot truly be assessed until model error is taken into account.

Line 25:

"When successful, it provides confidence that the method may be applied to real data. When it fails, it helps us figure out why."

Change to:

"This is a prerequisite to applying the method to real data."

Line 30:

The term "synchronization error" is reasonable when the observations are perfect.

However when the observations are noisy, I believe it's more common in DA to call the term SE the Root Mean Square Deviation (RMSD). If the observations were known perfectly then SE could also be called the Root Mean Square Error (RMSE). I should also point out that in DA the difference [y-x] is often called the 'innovation' within the context of the update procedure, and the "OMF" (for observed-minus-forecast) outside of that context.

In general, it is my preference to avoid the term "error" (e.g. SE or RMSE) when the observations are not known perfectly (i.e. in real-world experiments) but instead use RMSD. The word 'error' tends to give the wrong impression that a smaller value is better, which is not necessarily true when the observations are noisy.

Page 6:

Line 3:

"so that each state component's contribution to the synchronization error is weighted approximately equally."

It seems that the scaling applied to the observed variables could potentially be highly susceptible to noise or outliers. Is this the case?

Line 9:

"It is crucial to compare SE(t) for both estimates and predictions"

The DA terminology for that would be 'analyses' and 'forecasts', respectively.

Line 10:

"It is crucial to compare SE(t) for both estimates and predictions, as the former is just a 'fit' involving measured quantities, while the latter relies on accurate determination of the unmeasured variables as well."

I think it is worth mentioning again that the former is called "observation-minus-analysis

(OMA)" and the latter is called "observation-minus-forecast (OMF)" in operational applications. It is well understood in the atmospheric DA community that the OMFs are the more important measure for improving forecast skill, as OMAs can be set arbitrarily small by design.

Lines 12-14:

"Accurate estimates alone are not sufficient to validate the model or indicate the success of the estimation procedure, as they do not [contain] any information about the unobserved states."

They do contain information about the unobserved states from past observations propagated through the model from previous analysis cycles. The more important distinction is that the OMAs can be made arbitrarily small, but the observations contain errors and so this is not an indication of a better estimate. Rather, we would like to see consistent statistics of the OMFs over many cases showing a reduction in mean departure (assuming there are no biases in the observation errors).

Line 15:

"We have previously shown that when the synchronization error Eq. (9) decreases in time to very small values, the full state"

I don't think this can be true if there are large/outlier observation errors. At that point you would be fitting the noise and likely disrupting the state estimate and subsequent forecast. Do you mean the average SE decreases in time?

Section 4, Nonlinear Shallow Water Equations:

Line 26:

"We argue that the results presented here, for this simplified model, will be applicable for establishing the initial state of those models and predicting their subsequent behavior."

There is insufficient justification for that argument presented here. There are some additional experiments that must be done with this method before this claim can be stated confidently. A next step would be to apply the time-delay method an example case consisting of multiple layers while computing the temperature and salinity components of the density. This experiment would give a better test of estimating unobserved variables, though this is probably beyond the scope of this work.

Page 7:

Line 5:

I'm curious what impact the wind forcing had on the results. Have the authors tried experiments with multiple wind fields?

Line 7:

"With these fixed parameters the shallow water flow is chaotic, and the largest Lyapunov exponent for this flow is $\lambda\text{max} = 0.0325/\text{h} \approx 1/31$ h."

Please explain how the Lyapunov exponent was computed for this case.

Line 9:

You should clarify that the grid size {16,32,64} is changing resolution over the same domain boundaries.

This result is interesting, because it implies that given an observing network, you can determine the exact scale of features that can lead to synchronization by using that observing network. There is significant interest in transitioning climate models to higher resolutions going into the future (e.g. http://cpo.noaa.gov/ClimatePrograms/ModelingAnalysisPredictionsandProjections/OutreachPublications/MeetingsWorksh while at the same time a number of new satellites are coming on with much higher spatial resolution (e.g. SWOT Altimetry http://journals.ametsoc.org/doi/abs/10.1175/JTECH-D-13-00109.1). The authors

are encouraged to explore this concept in more depth.

Line 10:

"we estimated that approximately 70% of the D = 3Nˆ2 degrees of freedom must be observed in order to synchronize the model output with the data"

What observing network did you use? I.e. what was the distribution of observations? Were they stationary? Are they observed all at once, or throughout the observation window. These details will impact the number of observations needed.

Line 16:

"We are confident that despite the numerical challenges associated with scaling the algorithm up to larger D, the results presented here for N∆ = 16 will also remain valid for higher grid resolution."

Does this also hold if the grid resolution is kept constant but N∆ is increased by increasing the domain size? 50km grid spacing at the equator is about the resolution of many global ocean models run in operations. Do you expect a 1/2° global model to require around 70% coverage? A dynamic observing system tends to require fewer observations to maintain synchronization (e.g. see Penny, 2014, where the Lorenz-96 system is constrained by randomly located observations http://journals.ametsoc.org/doi/abs/10.1175/MWR-D-13-00131.1). A methodology for calcuating Ls for more realistic observing systems would be a valuable contribution.

Line 18:

"In the discussion above, which included reference to the lectures of [Cardinali] Cardinali (2013), we see that the requirement of having to observe[d] 70% of the model dynamical variables exceeds the measurements now available by at least a factor of two; more if the NWP model is larger [yet]."

Change to:

"In the discussion above, which included reference to the lectures of Cardinali (2013), we see that the requirement of having to observe 70% of the model dynamical variables exceeds the measurements now available by at least a factor of two; more if the NWP model is larger."

Page 8:

Section 5, Results with Time Delay Nudging for the Shallow Water Equations:

The equations on this page are missing equation numbers.

Line 9:

These look a bit like a stream function but usually that's written in the form: u = ∂Psi/∂y, v=-∂Psi/∂x

Just make sure this is what you intended.

Line 16:

Using a diagonal coupling matrix G ignores spatiotemporal correlations between points. Do you have any comments about the implications of using a diagonal G?

Lines 16-19:

I thought G was the time delay space coupling and g the physical space. But, here is says g is the time delay space coupling term. Is this a typo?

Line 17:

"These are chosen because the height values are several orders of magnitude larger than the flow velocities."

You could consider normalizing the innovations by the corresponding observation errors (i.e. standard deviation used for Gaussian noise).

Line 23:

Please cite a reference for calculating the average mutual information, using whatever method was used here.

Line 24:

"Furthermore, the results were reasonably stable to changing its value within a few [delta] t."

I'd like to see some analysis of the sensitivity to delta t to justify this statement.

Lines 26-27:

Do you use a time-delay extending before the beginning of the analysis cycle window [0,T]?

Page 9:

Line 3:

"Since DM $\geq$ 8 produces error values several orders of magnitude smaller than those obtained with DM $\leq$ 6, we expect the state estimates x(T) obtained with DM $\geq$ 8 to be quite accurate when compared with the estimates for DM $\leq$ 6. "

There seems to be a critical point between D_M=6 and D_M=8, what are the results for D_M=7? How do you explain this bifurcation? Is there a waveform that can only be resolved at this time delay length?

Line 11:

"Just a reminder note here, we used L = 256 = 33% of the total 768 dynamical variables as observed, then used time delay information on the waveform of the measurements to provide the required additional information."

This reminder is not necessary here in the results section. Instead, the clarification should be made earlier, perhaps when the value of Ls is stated on page 7 line 15.

Line 20:

"As this point is the key theme of this paper, we take the liberty of repeating that the number of physical measurements is just 33% of the overall dynamical variables."

Yes, but the height has a strong relationship with the currents. What kind of results do you get if you observe only the u or v components of the velocity instead of height? Is 33% still sufficient?

Line 26:

"in accordance with our previous results in Whartenby et al (2013)"

Change to:

"in accordance with Whartenby et al. (2013)"

Lines 31-32:

"When enough information is available, and the coupling is strong enough, these conditional Lyapunov exponents will all be negative, allowing the coupled systems of data and model output to synchronize."

Do you have a means of computing the conditional Lyapunov spectrum for this system? If so I would like to see these results presented.

Page 10:

Line 7:

"in a true experiment, the success of the assimilation procedure must be evaluated against the predictions - not the estimates."

Again, if you are going to discuss real-world problems, I suggest using the appropriate terminology: "forecasts - not the analyses."

Line 19:

"We remark, however, that the overall space of parameters appearing in our formalism

has not been thoroughly explored and that by further adjusting these parameters"

One obvious example to maintain prediction skill with reduced observations would be to use non-diagonal coupling terms in your nudging (as is standard for operational data assimilation).

Line 22:

"One would expect, that at some point the resolution should be high enough to not necessitate further measurements."

I don't understand this statement. In a realistic system, as the resolution increases, the resolved features also increase. In that case, I would expect the required observations to increase super-linearly. I suggest more experimentation should be done before making such a claim.

Section 5.1 Measurements with Gaussian Noise

Line 24:

"In operational data assimilation in meteorology, one challenge is that the measurement contains observation error."

This is true in all data assimilation, regardless of the domain.

Change to:

"In operational data assimilation, observations contain measurement errors and systematic biases."

Line 29 and 30:

Missing equation numbers

Line 31:

"and we selected $C\_data = 106$ and $C\_height = 1652$."

Please list the scale of the observation errors used in the experiments shown in figure 7 within the text.

Page 11:

Line 2-3:

"The time delay nudging method remains robust under imperfect observations."

I would anticipate there are observation errors that can cause outliers large enough to 'break' this method. Have the authors found such cases?

Section 6. Using Drifter Data with Time Delays

Line 9:

The authors should mention here if they append the drifter coordinates to the state vectors, or if they convert the drifter positions to an Eulerian velocity measurement.

Lines 16-19:

"After the initial deployment, the drifters move between grid points providing information not available from grid point measurements alone."

In order for this to be true, the drifter positions shouldn't be determined solely by the gridpoints. For example, they might be computed on a finer model grid. However in the next paragraph it is said:

"The dynamics of drifters is described as two-dimensional fluid parcel motion on the surface of the water layer. Since the positions of drifters are continuous values, the velocities of the drifters are estimated by a smooth linear interpolation"

So, if the velocities are determined by a smooth linear interpolation, then aren't they determined from the grid point measurements alone?

Line 28 and 30:

Missing equation numbers

Line 28 looks as if it should have a left bracket at the beginning of the line.

Page 12:

Perhaps you might also consider an additional case with L = 208 + 20 while N_D = 0 to ensure a fair comparison, or at least give more perspective on the impact of having a portion of the obs drifting versus stationary.

Lines 5-6:

"We have further investigated how the geographic distribution of the drifters influences the size of the synchronization error, although these results are not displayed here."

Why not? These results would be interesting.

Line 11:

Is it possible to generate an estimated current velocity based on the wind field and heights? Does that velocity correspond well to the drifter observations?

Section 7. Conclusion

Line 15:

"In an earlier paper Whartenby et al (2013) we showed that using standard nudging"

Change to:

"Whartenby et al. (2013) showed that using standard nudging"

Line 22:

"realistic and complex models of the ocean, atmosphere system"

Change to:

"realistic and complex models of the ocean [or] atmosphere"

Line 26:

This discussion about 4DVar should be a section either in the introduction or just after, giving the context of the time delay nudging method relative to the 4DVar method.

Page 13:

Most of the discussion on model error should be moved to the body of the manuscript. It does not seem appropriate in the conclusions.

Page 14:

Line 8:

"The framework presented here allows one to directly estimate the minimum number of observations at each measurement time required for accurate predictions, Ls"

I'm not sure I would characterize it as a direct method for estimating L_s. A brute force application of many values until achieving synchronization seems more indirect. A more direct method would be desirable.

Line 12:

Because 'almost surely' has a very specific mathematical meaning, I would suggest to change:

"real data will almost surely fail"

Change to:

"real data will likely fail"

Line 13:

"On the other hand, when the process succeeds, it increases confidence that predictive failures associated with the assimilation of real data arise from inadequacies in the model."

Or, inadequacies of the observing system.

Line 14:

"When the model is wrong, as it typically will be in practice"

Change to:

"When the model is imperfect, as will be the case in practice"

Line 16:

"In other words, when predictions fail, our strategy provides a useful diagnostic framework to help determine where to concentrate our efforts: improving the model or collecting more data."

This statement should be tempered or removed. I don't see such a tradeoff being made in a real-world scenario, in short because there are far more considerations that go into such decisions than whether one can get a better analysis at a given time. For better or worse, these efforts tend to be independent. There are obvious reasons to develop both the model and increase data collection in parallel. For example, model development can always be postponed, but data collection can never be repeated.

Lines 18-19:

"The inclusion of time delays comes of course with an additional computational cost, mainly associated with the integration steps required to construct the time delay vectors and its Jacobian, as well as solving for the perturbation itself."

This should be stated at the very beginning when the method is first introduced.

Line 24:

This should be the leading paragraph in the conclusion section.

Page 19:

Figure 2:

This is not a complete sentence: "In accordance with the synchronization error results."

---

## Referee Comment (RC2) · Anonymous Referee #2 · 24 May 2016

**Nonlinear Processes in Geophysics Discussion. Authors: Zhe An, Daniel Rey, JingXin Ye and Henry D.I. Abarbanel**

24 May, 2016

Title: *Estimating the State of a Geophysical System with Sparse Observations : Time Delay Methods to Achieve Accurate Initial States for Prediction*

Recommendation: **Minor Revision**

**General Comment**

This is a very interesting and relevant study, timely and central on one of the most appealing research topic in data assimilation nowadays.

The methods describes a new formulation of nudging using time delay observations. While the approach has been already presented in previous papers from some of the same authors, the present manuscript adds a relevant application to a dynamical system of larger dimension so making this study appealing in the perspective of applications to more realistic model-observation scenarios.

I suggest the manuscript to be accepted subject to a minor revision directed mainly to improve the readability and the presentation of the results. In fact in a few places the manuscript lacks of the necessary details to make it clearer to a slightly broader audience than just experts in data assimilation.

Find below specific comments that I would ask the authors to address during their revision.

**Specific Comment**

1. Abstract. The authors should not introduce in the abstract alone important quantities, such as the critical threshold, $L_s$. This must be re-introduced in the Introduction or at least somewhere else in the main body of the article. Note for instance that it is only in the abstract that it is stated clearly that the work deals with chaotic dynamics. While this is patent in the rest of the paper it must be said explicitly.

2. Introduction. Lines 15-17, page 1. The statement is strictly true only for chaotic systems (this relates to my previous comment).

3. Page 2, lines 7-11. When talking about the necessity to control the unstable modes, it is relevant to mention methods, like the Assimilation in the Unstable Subspace, in which the analysis update is explicitly designed for this purposes; (see *e.g.* Palatella, L., A., Carrassi and A. Trevisan, 2013. Lyapunov vectors and Assimilation in the Unstable Subspace: theory and applications. J. Phys. A: Math. Theor. 46, 254020 )

4. Page 2, line 12. The reference is Pazo et al., 2016 (Pazo, D., A. Carrassi and JM. Lopez, 2016. Data Assimilation by delay-coordinate nudging. Q. J. Roy. Meteor. Soc. 142, 12901299)

5. Page 2, line 16. $L_s$ is not properly defined. The authors should explain a bit more precisely what it is meant by "critical threshold" ?

6. Page 3, line 5. Notation or text must be improved. Is $\mathbf{y}$ a L-dimension vector? If so, you better state that you take $L$ observations that are then collected in the observation vector $\mathbf{y}$.

7. Page 3, line 13. Again notation or text must be improved. The equation $y_l(t) = x_l(t) + noise$ suggests that $L = D$ which is not the case in your experiments, and the fact that $L \ll D$ is indeed one of your key point. You should say that you use an operator $H$ that only observes a portion of the state-vector (mainly the heights in the experiments that follow).

8. Page 3, Eq.(2) and lines 24-30. Eq.(2) requires observations at each time-step of integration, something which is usually obtained by interpolating in between successive observations in real applications. For the sake of completeness, it must be also added that the condition of a negative conditional Lyapunov exponent implies the setting for the strength of the nudging forcing, **g** and not just on $L$. In classic literature in fact this is usually the case, and the observational network is given.

9. Page 4, Eq.(7). I think **g** and **G** should not be bold.

10. Page 7, line 15. How is $L_s$ obtained ? Does it come from the simple nudging case Eq.(2) ?

11. Page 7, line 18. "Cardinali" is written twice. Line 19: "observed" should be "observe".

12. Page 8, line 20. The sentence about parameter estimation relates to the chosen "perfect model" scenario. It would be better if the author states this clearly.

13. Page 8, line 22-24. You might want to say something more on this regard. How is it optimized? What does it mean average mutual information? Is it the time decorrelation scale?

14. Page 9, line 4-6. Are you computing the error using only the observed components of the state vector also for $t > T$?

15. Page 10, line 1. Do you mean Fig.5?

16. Page 10, line 5-6. Do you have suggestions on how to select the coupling?

17. Page 10, line 11-13. In fact the values chosen for $L$ are very close to each other and results highlight a strong sensitivity of your method to this (error diverge when $L = 248$). Can you comment more on this? Another aspect regards how those observations are placed. One can always achieve a better control of the error by a proper deployment of the observations (possibly with the use of target observations). What would it happen if observations were denser in the proximity of the most dynamically active areas?

18. Page 10, line 18-22. The final paragraph of Section 5 is important but it necessitates some improvements: (1) You might want to say that, as known from classical synchronization results, the optimal forcing strength (g and G in the present context) depends on the number and distribution of the observations; (2) the conclusion relating the model resolution and observations network is unclear. Even with a high resolution model one may still necessitate a growing number of observations to keep under control the unstable modes. Please clarify this point.

19. Section 5.1. A number of details are unclear in the present version of the Section. In particular one gets easily confused by the mix of information on how observations are made noisy given in the caption of Fig.7 and the fields "data" in lines 29-30. (1) What is $\phi$ in the first equation? Did you mean $\psi$?; (2) From the $2^{nd}$ equation one sees that the observations have zero mean, which seems inconsistent. (3) In the experiments so far you have only observed $h$, why are you then showing how observational noise is simulated for the velocities if the latter are not assimilated? (4) What are the values of $C_{height}$ and $C_{data}$ and how are they chosen? We do not know how these values scales with respect,

for instance, to the system climate variance in the same variables. It is consequently impossible to judge to which extent this observational error is small or big. (5) Please make consistent the text in Section 5.1 with the Fig. 7 caption. From the latter one learns how observations are being perturbed with normal distribution with variance $\sigma$.

20. Page 11, line 6. The reference should be Mariano et al (2002). Isn't?

21. Page 11, line 18-30. This part is very interesting but key details are missing on the state-augmentation formulation that the authors seem to have used to incorporate drifts. Please improve presentation on this aspect and provide more details.

22. Page 13, line 23-25. Although its meaning may be clear to a reader from the data assimilation community, $R_m$ is undefined? It may be convenient to define it in relation with Eq. (12), *i.e.* with $R_f$.

23. Page 12 - 13. The long discussion on the equivalence with smoother (4DVar) is very interesting but, in my opinion, very badly placed in the middle of the conclusion. That is not a conclusion, but rather a discussion. I think the authors should move it in the main body of the manuscript, perhaps when the method is presented and before the numerical results. In any case an independent section would be ideal.

24. Page 14, line 27-34. This comes too late and it would be better seen at the beginning of the Conclusion, when you recall the motivation behind the research effort.

25. Page 15, line 1. You have repeated already many times the model you have adopted.

26. Page 15, line 5-6 and line 13. Check the reference to Cardinali (2013).

27. References. Some entries in the list have typos that require corrections, particularly in the page information which has "?" instead of "-". This is the case, for instance, of Kuznetsov et al, 2003, but the problem is present elsewhere as well. Make consistent use of journal names abbreviations throughout all entries.

28. Figure 6. Correct labels and box in the top panels.

---

## Author Comment (AC2) · 7 Jun 2016

**Authors' Response to Referee**

May 27, 2016

**Overall**

Thanks for the suggestions from the referee. The paper will be better organized after we incorporate the modification based on the comment of the referee. The essential details will be added in the final version.

**Responses to Specific Comment**

5. Page 2, line 16. $Ls$ is not properly defined. The authors should explain a bit more precisely what it is meant by "critical threshold" ?

**Response:** In Whartenby et al's paper and this paper, different dimensions of the observation have been tested to achieve the accurate prediction. When the dimension is larger than the threshold $Ls$, the prediction quality is reasonably acceptable, and the root mean square error (RMSE) can be reduced to a relative low order. However, for any dimensions smaller than Ls, the RMSE increases abruptly, which indicates the approach to the accurate prediction fails.

(Whartenby, W., J. Quinn, and H. D. I. Abarbanel, "The Number of Required Observations in Data Assimilation for a Shallow Water Flow," 20 Monthly Weather Review 141, 2502-2518, (2013).

6. Page 3, line 5. Notation or text must be improved. Is **y** a L-dimension vector?

**Response:** **y** is an $L$ dimensional vector, which is constructed by the measurement on the $L$ grids.
* * *
7. Page 3, line 13. The equation $y_l(t) = x_l(t) + noise$ suggests that $L = D$ which is not the case in your experiments, and the fact that $L << D$ is indeed one of your key point. You should say that you use an operator H that only observes a portion of the state-vector (mainly the heights in the experiments that follow).

**Response:** As indicated in the paper, the index $l$ is in the range of $\{1, 2, ...L\}$. $L$ is the dimension of the observation state. The operator H is an identical matrix which is omitted.
* * *
10. Page 7, line 15. How is Ls obtained? Does it come from the simple nudging case Eq.(2) ?

**Response:** $Ls$ is the critical threshold of the simple nudging method. When the dimension of observation $L$ bigger than $Ls$, the synchronization will be achieved. As a benchmark of the traditional method, it has been compared with the sufficient dimension L in this study.
* * *
12. Page 8, line 20. The sentence about parameter estimation relates to the chosen " perfect model" scenario. It would be better if the author states this clearly.

**Response:** Perfect model scenario means the model is error free $(R_f = 0)$. Only the observation error is considered in the study here.

13. Page 8, line 22-24. You might want to say something more on this regard. How is it optimized? What does it mean average mutual information? Is it the time decorrelation scale?

**Response:** Average mutual information in the delay embedding theorem is used for determining the embedding delay parameter. From the average mutual information, the independent coordinate can be constructed* **.

* Abarbanel, Henry D. I., Analysis of Observed Chaotic Data, Springer-New York (1996)
** Rey, Daniel, Michael Eldridge, Mark Kostuk, Henry D. I. Abarbanel, Jan Schumann-Bischoff, and Ulrich Parlitz, "Accurate State and Parameter Estimation in Nonlinear Systems with Sparse Observations", Physics Letters A 378, 869-873.(2014)..

14. Page 9, line 4-6. Are you computing the error using only the observed components of the state vector also for $t > T$?

**Response:** In this paragraph, the error in the prediction window $(t > T)$ is computed by the observed components (h). In the following analysis, the error is computed by the unobserved components (u, v).

Since all information (observed and unobserved variables) is known in the twin experiment, The errors of the unobserved components are also the criteria of a valid assimilation.

16. Page 10, line 5-6. Do you have suggestions on how to select the coupling?

**Response:** The coupling strength is tested by a continuous increment. The results displayed here are compared with the results g = 0.5 and g = 1.5. As

discussed above, when the conditional Lyapunov exponent is negative, the synchronization will be achieved.
* * *
17. Page 10, line 11-13. In fact the values chosen for $L$ are very close to each other and results highlight a strong sensitivity of your method to this (error diverge when L = 248). Can you comment more on this? Another aspect regards how those observations are placed. One can always achieve a better control of the error by a proper deployment of the observations (possibly with the use of target observations). What would it happen if observations were denser in the proximity of the most dynamically active areas?

**Response:** In this study, the variations of variables (after normalization) at different grids are in the same order. There is no "more dynamically active areas" in this simulation. Moreover, the observation locations are randomly chosen on the grids. As the referee's comment, it's the fact that a proper deployment of observations will induce a higher prediction quantity. This may be included in our future study.
* * *
18. (2) the conclusion relating the model resolution and observations network is unclear. Even with a high resolution model one may still necessitate a growing number of observations to keep under control the unstable modes. Please clarify this point.

**Response:** In the study, the resolutions 16x16, 32x32 and 64x64 were tested, we found the proportion of the necessary observation L to the dimension of the system D will reduce in some systems with higher resolutions. While, the conclusion that the necessary dimension of time delayed nudging method will be much less than the one in the traditional nudging method always holds.
* * *
19 (3) In the experiments so far you have only observed h, why are you then showing how observational noise is simulated for the velocities if the latter are not assimilated? (4) What are the values of $C_{height}$ and $C_{data}$ and how are they chosen? We do not know how these values scales with respect.

**Response:** (3) Those are not observational noise for velocities. This is the initialization error of the stream function. (4) $C_{height}$ and $C_{data}$ are empirical choices which are in the rational scale of observation error. The assimilation results are not very sensitive to the choices of these parameters.

**General Technical Corrections**

Other questions focus on the rephrasing, referencing and formatting. We will make the changes and polish the paper in the revision of our paper. Thanks for the thorough reading and detailed suggestions.

---

## Author Response (AR1)

**Authors' Response to Referee, Dr. S.G. Penny**

First, may we thank the referee for the thorough reading of our paper and for the detailed suggestions of changes and improvements. This document is our first response to your comments, and when the discussion period for the paper is completed, we will incorporate our comments here and responses to all commenters into a revised version of the paper. Again, these remarks have been most helpful.

The Authors: An, Rey, Ye and Abarbanel

\_\_\_\_\_

 Techniques for dealing with a sparse observational networks are critically important, particularly for ocean and climate reanalyses that attempt to reconstruct the past state of the Earth system (e.g. Compo et al., 2011; http://www.esrl.noaa.gov/psd/data/gridded/data.20thC\_ReanV2.html). The experiment scenarios described here by the authors are perhaps most applicable to the estimation of the global ocean state after the introduction of satellite altimeters, e.g. TOPEX/Poseidon in late 1992 (https://sealevel.jpl.nasa.gov/missions/topex/), with their final set of experiments having a potential application to leverage data from the Global Drifter Program (http://www.aoml.noaa.gov/phod/dac/index.php). Thus from a practical point of view, the time-delay method has potential merit for operational scale data assimilation (DA) and reanalysis.

\_\_\_\_\_

We have examined the links you provide, and it does indeed look like fruitful directions for the use of the time delay method. We think it fair to evaluate ourselves critically and recognize that we may not be prepared to tackle, with present personnel levels and computational resources, something of this magnitude. However, we agree about the importance of this problem and appreciate your encouraging comments and suggested applications. There is no doubt in our mind that these items are in our future, and we look forward to pursuing them.

\_\_\_\_\_

2. Because of such potential, the authors should give a bit more explicit description about how these ideas compare to common methods like 4DVar or the 4D Ensemble Kalman Filter (EnKF), both of which utilize observations over an extended time window. The authors could give a more thorough depiction of how their ideas could be incorporated in these existing systems in order to facilitate a higher likelihood that an operational center might adopt the approach.

\_\_\_\_\_

Again, we agree in toto with the referee's comments. We have taken a more cautious path to these comparisons with ExtKF, EnKF, and traditional 4DVar methods [but see our paper in NPG (Improved variational methods in statistical data assimilation J. Ye, N. Kadakia, P. J. Rozdeba, H. D. I. **Abarbanel** and J. C. Quinn Nonlin. Processes Geophys., 22, 205-213, doi:10.5194/npg-22-205-2015, 2015))].

The comparisons require, in our opinion, another paper dedicated to those, and, if we want to be fair about the comparisons, we feel we need to do them in cooperation with colleagues who have experience with those methods. We do have such colleagues at Scripps Institution of Oceanography, and we will be working with them on just these matters.

We considered going into more detail in this paper and decided it might take away from its main point, which was to demonstrate the benefit of using time delays in a simple geophysical model, and its application to drifter measurements.

\_\_\_\_\_

\_\_\_\_\_

3. The sea surface height is closely connected to the near surface currents via the geostrophic balance, particularly in midlatitudes. Thus it is expected that unobserved currents would be well constrained by proper estimation of the surface height. For example, sea surface heights and sea surface winds are used to construct an estimate of ocean surface currents for the OSCAR product (http://www.oscar.noaa.gov/index.html). However, the examples given by the authors could perhaps be described as a supplement for the tropical region where this relationship breaks down. For future work, a natural extension would be to address a slightly more sophisticated example consisting of multiple vertical layers and the modeling of the temperature and salinity components of the density. This experiment would give a better test of estimating unobserved variables. For example, observing only temperature while estimating salinity is a challenging problem for ocean reanalyses before the Argo era.

Thank you for these suggestions. We will look closely at how the geostrophic balance plays into directing the dynamical outcome of our use of time-delays. However, as the geostrophic wind is related to the gradient of the height variable, it may be that this provides a different general constraint on the solutions to the shallow water equations.

4. A brief statement could be made about the applicability of the time-delay approach, for example, to the tropical observing system of moored buoys (TAO/TRITON). These are stationary sensors generating data about once every 10 minutes, but the majority of this data is not used in DA because most global scale ocean assimilation systems use analysis cycles that span multiple days. Even a coupled ocean/atmosphere DA system cycling every 6 hours could benefit from better use of this data. I suggest investigating the TPOS-2020 (Tropical Pacific Observing System) effort for the potential to inform the future development of this and other observing systems (http://tpos2020.org). A weakness in the chosen experiments scenarios that should be acknowledged is that the approach has not been tested on time-delay observations with errors that are correlated in the time dimension. This is particularly important in ocean DA because errors of representativeness often dominate (versus instrument errors).

This is an excellent suggestion. It would appear to provide information from an unused (by us, and it appears many others) data source for useful information about ocean models.

- - 5. I suggest an experiment, perhaps for future work, in which you run 2 model resolutions. The high resolution run is treated as 'truth', from which observations are drawn. The low resolution model is what you are synchronizing via DA. Set up appropriately, this should give you 'natural' errors of representativeness in the observations that may be correlated in time with the errors of future or past observations. Does the time-delay method still work effectively in this experiment scenario?

\_\_\_\_\_

\_\_\_\_\_

This is another good idea. We actually considered including such an experiment, to investigate the impact of finite resolution and model errors arising from subgrid scale processes. Ultimately, we decided to leave these considerations for a future paper, and focus here on the perfect model scenario. We see no impediment to the use of time delays in this scenario; indeed, it may provide information from "spatial delays" (also used in the past for nonlinear dynamical descriptions of waves propagating in nonlinear materials) presently not incorporated in our own work.

\_\_\_\_\_

6. The time-delay method is described in comparison to nudging as a baseline. I would like to see the authors compare a simple 4DVar to the time-delay method as well (via experiment) to give context into how their method compares to a more state-of-the-art DA.

A thorough comparison is planned for a future paper, where we discuss in detail the connection between our method and 4DVar. The revised paper we will prepare notes this as future work. The simple answer at this time is that we know how to introduce time delays into what we call the action, often called the 4DVar cost function, and we have not yet used this augmented cost function (and our method of 4DVar as noted above) on this problem.

7. It seems that the time-delay information for the observations and model state applied with what is essentially a diagonal coupling term emulates a similar effect as the cross-covariances that would in effect apply a non-diagonal coupling term to the innovations computed at different times throughout the window. The authors should discuss how the off-diagonal coupling used in most operational DA relates to the diagonal coupling with time-delay observations used here. We agree with your statements here about the cross-correlations. The off-diagonal terms here arise from the generalized inverse of the time delayed innovations. The diagonal coupling term in time delay space could for instance damp the effect of measurements further in the future, which have more uncertainty due to dynamical instability.

A similar effect could be achieved from 4DVar with a uniform prior and a time distributed observation error matrix, but we would rather discuss this in a future paper that more thoroughly explores the connection between time delayed nudging and 4DVar.

\_\_\_\_\_

8. The impact of observation error on synchronization via the nudging approach is not addressed very thoroughly. I'd like to see some evaluation of the sensitivity to observation error in the assessment of the method. The authors should describe how their method is impacted by outliers in the observed data. Is the method sensitive to such outliers? I'd like to see an example.

When observation error is present, the model will synchronize to within the noise ball of the `true' solution, when the model is known perfectly and enough observations are present. We recognize however that for many DA methods the goal is to reduce the RMSE below the noise level, but this was not the case here as we chose to consider the sparsity of observations as the dominant effect, rather than observational noise. As a result, we elected to only include a brief investigation, to show that our method is not significantly impacted by very small observational errors.

To be clear though, you are right that enough noise will 'break' this method, or at least severely impede its chances of success. The degree of regularization needed for the generalized inverse of dS/dx is commensurate with the observational errors of the system.

In addition to these remarks, we use the synchronization error as our "monitor" of the reduction of the model output error to indicate when we have sufficient observations at each measurement time. These errors are limited by the noise in the observations.

General Technical Corrections:

We do not comment on these, really valuable—to us—comments. We have addressed each of them in our rewrite of the submitted paper, and on revision after the end of the NPG discussion period, we will note each change we have made built upon these detailed, and appreciated, comments. Thank you.

**Estimating the State of a Geophysical System with Sparse Observations : Time Delay Methods to Achieve Accurate Initial States for Prediction**

Zhe An1, Daniel Rey1, JingXin Ye1, and Henry D. I. Abarbanel1, 2

1Department of Physics, University of California, San Diego, 9500 Gilman Drive, La Jolla, CA 92093-0374, USA 2Marine Physical Laboratory (Scripps Institution of Oceanography) University of California, San Diego 9500 Gilman Drive La Jolla, CA 92093-0374, USA

Correspondence to: Zhe An (z1an@ucsd.edu )

Abstract. The problem of forecasting the behavior of a complex dynamical system through analysis of observational timeseries data becomes difficult when the system expresses chaotic behavior and the measurements are sparse, in both space and/or time. Despite the fact that this situation is quite typical across many fields, including numerical weather prediction, the issue of whether the available observations are 'sufficient' for generating successful forecasts is still not well-understood.

- 5 An analysis by Whartenby et al (2013) found that in the context of the nonlinear shallow water equations on a  $\beta$ -plane, standard nudging techniques require observing approximately 70% of the full set of state variables. Here we examine the same system using a method introduced by Rey et al (2014a), which generalizes standard nudging methods to utilize time delayed measurements. We show that in certain circumstances, it provides a sizable reduction in the number of observations required to construct accurate estimates and high-quality predictions. In particular, we find that this estimate of 70% can be reduced to
- 10 about 33% using time delays, and even further if Lagrangian drifter locations are also used as measurements..

**1 Introduction**

The ability to forecast the complex behavior of global circulation in the coupled Earth system lies at the core of modern numerical weather prediction (NWP) efforts. To successfully predict such behavior requires both a good model of the underlying physical processes as well as an accurate estimate of the state of the model at the end of the analysis or observation window.

15 When the model is chaotic, even if it is known precisely, the accuracy of the prediction depends on the accuracy of the initial state estimate. This is due to sensitive dependence to the initial conditions, which was first identified by Lorenz (1963).

Here we consider an idealized situation where a perfect dynamical model describes the deterministic time evolution of a set of D state variables. We also assume a set of L measurements are taken at each observation time from the physical system at a uniform sampling interval  $\Delta t$ , which is assumed to be small relative to the time scale of the dynamics.

Our main concern here is the case where the measurements are sparse in state space, so  $(L \ll D)$ . Although this discussion will focus solely on a specific geophysical system (the shallow water equations), the methods we describe here have broad applicability across the quantitative study of the underlying physical or biological properties appearing in many complex systems. In particular, the situation of high dimensional dynamics and sparse measurements is typical in the process of examining the consistency of observed data and quantitative models of complex nonlinear systems: from functional nervous systems to genetic transcription dynamics, among many other examples (Abarbanel (2013)).

As discussed by Cardinali (2013), operational NWP models at the European Centre for Medium-Range Weather Forecasting 5 (ECMWF) now contain upwards of  $10^8$  degrees of freedom. These models are analyzed using  $3 - 4 \times 10^7$  daily observations, a large portion of which are often discarded. Given the scale of these calculations, the question of whether the remaining observations are in fact 'sufficient' for producing accurate analyses and forecasts is of the utmost importance.

To clarify the term 'sufficient' we refer to the analysis by Whartenby et al (2013), which showed that familiar nudging methods, when applied to a chaotic, shallow water flow on a  $\beta$ -plane driven by Ekman pumping, require observation of

10 roughly 70% of the  $3N_{\Delta}^2$  dynamical variables. That is, to achieve accurate forecasts, these methods required measuring the height variable h and at least one of the two velocity variable u, v at each of the  $N_{\Delta} \times N_{\Delta}$  grid points. Additionally, the prediction accuracy was shown to drop precipitously when the number of observations L drops below a critical threshold  $L_s$ , which was identified as the number of 'required' observations.

The existence of this critical threshold, despite the otherwise ideal circumstances, raises a number of questions. Most notably 15 this one: what can be done if the number of observations L is constrained to be less than  $L_s$ ? In examining this question, it is worth considering that the value  $L_s$  depends on a number of factors, including the chosen data assimilation algorithm. This fact suggests that one might be able to effectively reduce this threshold by modifying the algorithm in a way that more efficiently utilizes the information in the available observations.

[revised manuscript text omitted]

- 15 Also worth mentioning is that here we are not using time 'delays' but rather a *time advanced* formulation, which looks forward in time. The reason for this is related to the goal of controlling the propagation of errors on the unstable manifold as the system is integrated forward in time, which are locally described by Eq. (3) so the time advanced construction is a natural choice, although both formulations are acceptable. This also brings up a concern regarding what to do at the end of the assimilation window. One option is to switch to a time delayed formulation, or perhaps a mixed formulation that uses delays
- 20 both forwards and backwards in time. Though comparing the performance of various choices would likely be interesting, we do not consider this issue further. Rather, our numerical experiments use only a time advanced formulation, by choosing the end of the observation window so that the last observation  $\mathbf{y}(T + \tau (D_M 1))$  is always available.

There appear to be considerable similarities between this method and those of strong constraint 4DVar (Lewis et al, (1985); Talagrand et al (1987)), which are now standard practice in data assimilation Rabier et al (2000). Although in this form,
time delay nudging certainly cannot handle a system of the size used in operational NWP, as the variational equation requires manipulating *D* × *D* matrices, it may be possible to avoid this issue, for instance by using adjoints, similar to what is done in practice (Courtier et al (1994)). This formulation will be given in a subsequent paper.

Pazo (2016) gave a simplified version of time delay nudging that requires considerably less computation. Although their approach has not yet been applied to geophysical flows, it is worth investigating whether it is also capable of achieving the same reduction in  $L_s$ .

Furthermore, while it known that chaotic behavior in the model can cause serious issues for strong constraint 4DVar (Pires et al (1996)), perhaps less well-known is that similar observability thresholds have also been observed in both nudging and 4DVar (Abarbanel et al (2009); Abarbanel (2013)), even with weak constraints (Quinn & Abarbanel (2010)). In fact, the value

of  $L_s$  appears remarkably consistent across a variety of formulations, which it may be a rather fundamental quantity. It is also evident that Kalman related methods can do better if the observation operator is allowed to adapt to the unstable subspace (Law et al (2014)). This is the related to the fact that the Riccati equation for the error covariance propagation targets the unstable subspace (Trevisan & Palatella (2011); Gurumoorthy et al (2015)). Related ideas have appeared in the literature on assimilation in the unstable subspace (Trevisan et al (2010); Palatella et al (2013)).

We are currently working on unifying the motivating ideas behind time delay nudging with the variational action principle of weak constraint 4DVar. This and other related connections to 4DVar will be given in a subsequent paper that will also compare time delay method with a few other common data assimilation techniques.

For the moment however, no additional theory will be introduced. Instead, we focus its application to a core geophysical model. Namely, the shallow water equations.

**3** Twin Experiments**

5

15

We test our time delay nudging procedure through a series *twin experiments* (Durand et al (2002); Blum et al (2009); Blum (2010)). After solving the original dynamical equations Eq. (1) forward from a preselected initial condition  $\mathbf{x}(0)$ , the observed data is taken as the projection down to the *L* observed components. Gaussian noise  $N(0,\sigma)$  is added to each component to simulate observation error.

To simulate the conditions of a *true experiment* we monitor our progress by calculating the observable synchronization error, namely the root mean square deviation between the data and the observed model states,

$$SE(t_n) = \sqrt{\frac{1}{L} \left| \mathbf{H} \cdot \mathbf{x}^s(t_n) - \mathbf{y}^s(t_n) \right|^2},\tag{9}$$

where scaled variables have been introduced such that  $x_{\ell}^{s}(t) = [x_{\ell}(t) - x_{\ell}^{min}(t)]/[x_{\ell}^{max}(t) - x_{\ell}^{min}(t)]$  and  $x_{\ell}^{min/max}(t)$  are the 20 minimum or maximum values of  $x_{\ell}(t)$  over the entire assimilation window. The same definition holds for  $y_{\ell}^{s}(t)$ . This rescales all data and observed model states to lie in the interval [0,1], so that each state component's contribution to the synchronization error is roughly equal. While this could make the result sensitive to outliers in the data, it did not appear to be an issue here.

When the estimation is completed at time t = T the coupling terms g(t) and G(t) are set to zero, and the uncoupled dynamics Eq. (1) are integrated forward from the estimated  $\mathbf{x}(T)$  to construct a forecast for t > T, which may then be compared with

additional observations y(t > T). Comparing against the forecast provides confidence that the unobserved state variables are also accurately estimated.

It was previously shown by Whartenby et al (2013) that when the synchronization error Eq. (9) decreases to very small values, the full state  $\mathbf{x}(T)$  is accurately estimated and the forecast is quite good. Conversely, when this fails to occur, the full state  $\mathbf{x}(T)$  is not well estimated and the prediction is unreliable.

In Rey et al (2014a, b), this contraction of the synchronization error was only observed when the number of time delayed observations  $L \times D_M$ , and the magnitude of the coupling matrices  $\mathbf{g}(t)$ ,  $\mathbf{G}(t)$  were 'large enough'. The precise meaning of this statement will become apparent shortly. We now describe the application of time delay nudging to a nonlinear model of shallow water flow on a mid-latitude  $\beta$ -plane. This geophysical fluid dynamical model (previously examined by Pedlosky (1987) and Whartenby et al (2013), among many others) is at the core of earth system flows used in NWP. Of course, operational models contain considerably more detail than

5 this example, and those models also describe the dynamics over a sphere. However, we suspect the results presented here for this simplified model will be applicable to more complex models as well, though we do not underestimate the numerical challenges in this extrapolation.

As the depth of the coupled atmosphere ocean fluid layer (10 - 15 km) is markedly less than the earth's radius (6400 km), the shallow water equations for two dimensional flow provide a good approximation to the fluid dynamics of the ocean. Three

10

20

fields on a mid-latitude plane describe the fluid flow  $\{u(\mathbf{r},t), v(\mathbf{r},t), h(\mathbf{r},t)\}$ : the north-south velocity  $v(\mathbf{r},t)$ , the east-west velocity  $u(\mathbf{r},t)$ , and the height of the fluid  $h(\mathbf{r},t)$ , with  $\mathbf{r} = \{x,y\}$ . The fluid is taken as a single, constant density layer and is driven by wind stress  $\tau(\mathbf{r},t)$  at the surface  $z = h(\mathbf{r},t)$  through an Ekman layer. These physical processes satisfy the following dynamical equations with  $\mathbf{u}(\mathbf{r},t) = \{u(\mathbf{r},t), v(\mathbf{r},t)\}$ ,

$$\frac{\partial \mathbf{u}(\mathbf{r},t)}{\partial t} = -\mathbf{u}(\mathbf{r},t) \cdot \nabla \mathbf{u}(\mathbf{r},t) - g \nabla h(\mathbf{r},t) + \mathbf{u}(r,t) \times (f(y)\,\hat{\mathbf{z}}) + A \nabla^2 \mathbf{u}(\mathbf{r},t) - \epsilon \,\mathbf{u}(\mathbf{r},t)$$

$$\quad \frac{\partial h(\mathbf{r},t)}{\partial t} = -\nabla \cdot \left[h(\mathbf{r},t)\,u(\mathbf{r},t)\right] - \hat{z} \cdot \operatorname{curl}\left[\frac{\tau(\mathbf{r},t)}{f(y)}\right].$$
(10)

The Coriolis force is linearized about the equator  $f(y) = f_0 + \beta y$  and the wind-stress profile is selected to be  $\tau(\mathbf{r},t) = \{[F/\rho] \cos(2\pi y), 0\}$ . The parameter A represents the viscosity in the shallow water layer,  $\epsilon$  is Rayleigh friction and  $\hat{z}$  is the unit vector in the z-direction. The values we have used for the model parameters are given in Table 1. With these fixed parameters the shallow water flow is chaotic, and the largest Lyapunov exponent for this flow is estimated to be  $\lambda_{max} = 0.0325/h \approx 1/31 h$ , measuring the average growth rate of random perturbations.

We have analyzed this flow using the enstrophy conserving discretization scheme given by Sadourny (1975) on a grid of size  $N_{\Delta}^2$  for  $N_{\Delta} = \{16, 32, 64\}$  with periodic boundary conditions. Using the twin-experiment framework, with simple nudging given in Eq. (2) and a static observation operator, approximately 70% of the  $D = 3N_{\Delta}^2$  degrees of freedom must be observed in order to synchronize the model output with the data (Whartenby et al (2013)). As the results are consistent across the various

25 grid sizes that were investigated, we restrict our discussion here to the case where  $N_{\Delta} = 16$ , so that the total number of degrees of freedom  $D = 3N_{\Delta}^2 = 768$ . For this case, Whartenby et al (2013) estimated  $L_s \approx 524 = 0.68 D$  using a uniform grid of observations. In other words, the height field and one of the velocity fields at each grid point needed to be observed to achieve reliable results.

Based on the discussion above and the lectures by Cardinali (2013), we see that the requirement of having to observe
70% of the model dynamical variables exceeds the measurements now available by at least a factor of two. This requirement is expected to be higher in practice, when the model and observations contain substantial errors. Furthermore, it is worth investigating whether the number of required observations eventually stabilizes to some finite value as the model resolution increases, but this is left for a future investigation.

| Parameter  | Physical Quantity                               | Value in Twin Experiments                         |
|------------|-------------------------------------------------|---------------------------------------------------|
| $\Delta t$ | Time Step                                       | 36 s                                              |
| $\Delta X$ | East-West Grid Spacing                          | 50 km                                             |
| $\Delta Y$ | North-South Grid Spacing                        | 50 km                                             |
| $H_0$      | Equilibrium Depth                               | 5.1 km                                            |
| $f_0$      | Central value of the Coriolis parameter         | $5 \times 10^{-5} \text{ s}^{-1}$                 |
| β          | Meridional derivative of the Coriolis parameter | $2.0 \times 10^{-11} \text{m}^{-1} \text{s}^{-1}$ |
| F/ ho      | Wind Stress                                     | $0.2 \text{ m}^2 \text{s}^{-3}$                   |
| A          | Effective Viscosity                             | $10^{-4} \text{ m}^2 \text{s}^{-1}$               |
| $\epsilon$ | Rayleigh Friction                               | $2 \times 10^{-8} \text{ 
[revised manuscript text omitted]

- In contrast to the previous results, where the initial conditions for the grid variables were taken to differ in both phase and 20 frequency between the true solution and the estimate, here the initial conditions only vary in amplitude. That is, the initial conditions of the data  $\psi_{data}(\mathbf{r}^{(i,j)},t_0)$  and  $h_{data}(\mathbf{r}^{(i,j)},t_0)$  and of the model  $\psi_{model}(\mathbf{r}^{(i,j)},t_0)$  and  $h_{model}(\mathbf{r}^{(i,j)},t_0)$  are related by  $\psi_{data}(\mathbf{r}^{(i,j)},t_0) = C_0 \psi_{model}(\mathbf{r}^{(i,j)},t_0)$  and  $h_{data}(\mathbf{r}^{(i,j)},t_0) = C_0 h_{model}(\mathbf{r}^{(i,j)},t_0)$ . We choose  $C_0 = 1.0 + 0.1 \eta$ , with  $\eta$ selected from a uniform distribution in the interval [-1,1]. The velocity fields are found as above, using  $\psi(\mathbf{r},t_0)$  as a stream function. This was done in order to improve the results, as we found that the drifter results were more sensitive to the choice 25 of initial condition than the results from the previous section, without drifters. Plots showing the initial positions of drifters for
- the two cases considered below ( $N_D = 20$  and  $N_D = 64$ ) are shown in Figure 8. They were also deactivated when they reached the boundary of the grid, so the number of operational drifters decreases throughout the estimation window.

In Figure 9, we show the synchronization error of observed quantities when for DM = 8, keeping all other parameters the same as in the previous calculations. We present (in red) the synchronization error for L = 208 height observations and
30 ND = 20 drifter observations, and we show (in blue) the same synchronization error when L = 208 and ND = 0 drifters are

deployed. With L = 208, namely, observing 27% of the heights and 20 drifters, the synchronization error converges to a small value within the five hour observation window. Without drifters, the estimation fails.

Furthermore, by increasing the number of drifters to  $N_D = 64$  within a 30 minute observation window, synchronization can be achieved with L = 128 height observations. Snapshots of the fields at different times throughout the estimation and prediction window are shown in Figure 10 for comparison.

Thus, although we have not yet explored how to balance between the number of drifters tracked and the number of height (or other) measurements employed, it is clear from these preliminary results that drifter data can be useful for improving the observability of the system, and that the time delay method provides a way to incorporate this information into the analysis.

**6 Discussion and Summary**

5

- 10 The transfer of information from measurements of a chaotic dynamical system to a quantitative model of the system is impeded when the number of measurements at each measurement time is below an approximate threshold  $L_s$ , which can be established in a twin experiment. Whattenby et al (2013) previously showed that for a nonlinear model for shallow water flow, a standard nudging technique given by Eq. (2) requires direct observation of roughly 70% of the dynamical variables  $\{h(\mathbf{r},t), u(\mathbf{r},t), v(\mathbf{r},t)\}$  at each measurement time to synchronize the model output with the observations.
- 15 Here we have demonstrated how information in the time delays of the observations may be used to reduce this requirement to about 30%, in which only the height fields need be observed. Moreover, it appears  $L_s$  can be even further reduced by adding positional information from drifters, which interpolate the height field at locations between grid points.

Although all this has been done on a simplified model of shallow water flow, implemented with only  $D = 3N_{\Delta}^2 = 768$  degrees of freedom, the process can be used to analyze increasingly realistic and complex models of coupled earth systems.

20 Since the successful analysis of simulated data is typically a prerequisite for success with real data, when the model is wrong (as it generally will be in practice) this methodology provides some idea as to whether the model is at fault, or whether more observations are needed.

Furthermore, we expect that this formalism will generalize to systems substantially larger than the one presented here, although we do not underestimate the numerical challenges involved in its extension to say, the scale of operational NWP

25 models. We also suspect this issue of insufficient measurements to be a critical limitation in our current ability to predict the behavior of complex, chaotic systems. Since such systems are quite typical in practice, these issues need to be examined with more realistic models.

Harking back to the introduction, we note that in the report by Cardinali (2013) indicates that 30-40 million daily observations are now available at the ECMWF, and that many NWP models comprise upwards of  $10^8$  degrees of freedom. If the

30 qualitative trends shown here, in which time delays provide successful predictions with only 30% of the state variables observed, can be extended to substantially larger systems, then this method may indeed be useful for improving the forecasts of existing operational NWP models. *Acknowledgements.* This work was funded in part under a grant from the US National Science Foundation (PHY-0961153). Partial support from the Department of Energy CSGF program (DE-FG02-97ER25308) for D. Rey is appreciated. Partial support from the MURI Program (N00014-13-1-0205) sponsored by the Office of Naval Research is also acknowledged. We would also like to thank the reviewers for their thorough reading and thoughtful suggestions that have helped significantly improve this manuscript.

[revised manuscript text omitted]

---

## Referee Report (RR1)

**Nonlinear Processes in Geophysics Discussion. Authors: Zhe An, Daniel Rey, JingXin Ye and Henry D.I. Abarbanel**

22 July, 2016

Title: *Estimating the State of a Geophysical System with Sparse Observations : Time Delay Methods to Achieve Accurate Initial States for Prediction*

Recommendation: **Minor Revision**

**General Comment**

I think the authors have overall well addressed my original concerns and have indeed substantially modified and improved the manuscript also in the light of the other Reviewer' comments.

I have only a few additional minor remarks for the Authors to address before acceptance. The comments that follow refer to the last Authors' reply and the numbering is made accordingly.

**Specific Comment**

- **Abstract**.
  I understand that $L_s$ is a lower boundary, so that synchronization is achieved when $L \geq L_s$. If so please add this piece of information in the text since the name critical number does not fully explain that it is also a minimum number. Also, please explain what the authors mean by synchronization here; a word explanation will suffice.

- **On point (4)**.
  The paper is still wrongly cited, as Pazo (2016) instead of Pazo *et al.* (2016), in line 7 page 6.

- **On point (18)**.
  I could not find where these statements are given in the current version of the manuscript. Maybe they have been removed, which is fine with me. Please clarify this and point the exact portion of the text. If the authors refer to the sentences between lines 4-7 at page 9, those are fine with me but I do not straightforwardly see how they relate to the statements originally under discussion.

- **On point (19)**.
  Here as well, I have problems to identify how and where in the text the authors have introduced modifications, if wished, to address my points. For instance, the current Section 5.1 does not seem to be related to the original Section 5.1, instead this seems to be the current Section 5.4. This makes me difficult to judge the changes. Regarding points (1) and (2), I could not find the equations in the current version of the manuscript. Are they still shown somewhere? As for my point (3), I referred to an apparent inconsistency between the type of observations you used. From everywhere in the text is clear you are working with height measurements, but from the equation in the original Section 5.1, it seemed you also create synthetic observations of the velocity. This issue does not seem to be present anymore in the current version (Section 5.4). Please indicate where in the text are placed the corrections mentioned in points (4) and (5).

- **On point (23)**.
  Please provide information on which actions, if any, you have taken to address this remark. By looking at the current version of the conclusion, it seems that all discussion on equivalence with the smoothers has been removed. Has it been placed elsewhere or modified somehow? Please clarify.

---

## Author Response (AR2)

Editor Decision: Reconsider after major revisions (further review by Editor and Referees) (30 Jun 2016) by Dr. Zoltan Toth

Comments to the Author: Dear Authors, Thank you for your original submission and the revised version of your manuscript. We received two very informative reviews. Both reviewers found your work worthy of publication but raised important questions and concerns about your manuscript. Given the substance and extensive nature of these reviews I will ask the two reviewers to look at your manuscript again once the revision is complete. However, closely inspecting the two reviews and your replies I found that further revisions are necessary before I can request comments from the reviewers again.

One concern is the selective and incomplete nature of your response to the reviewers comments. You chose to respond only to a few of the reviewers' comments while leaving most comments unanswered. Furthermore, in the cases where you did give a response in your reply, you did not indicate whether and how you responded to their questions and comments in the manuscript - except in the cases where you stated you chose not to make corresponding changes in the manuscript.

Specific comments related to your reply to Rev. 1's review:

Your item #2 - I agree with Rev. 1 and Rev. 2's related comment #23 and respectfully ask that you put your time-delay extended nudging technique in the context of existing 4-dvar methods. What is functionally common, what is different, how the two approaches differ conceptually and from theoretical considerations. This will enhance your manuscript and make it more accessible to the readers.

We have considerably revised the material related to the connection with 4DVar and added some additional details at the end of section two. A more thorough comparison will be done in a future paper.

Your item #4 - I echo the Rev.'s request for an explicit statement in the manuscript about a limitation of your study, namely that you have not tested the method with more realistic temporally correlated errors.

Yes we have added an explicit statement at the end of the introduction.

Your items #6 & #7 - Again, I find Rev. 1's suggestion helpful. A discussion of the relationship of your method and more commonly used approaches in the context of diagonal only vs non-diagonal coupling will be helpful for a proper assessment of the methodology you use.

We have included some comments on the use of diagonal vs non-diagonal coupling in our response to the referees. But they were not included in the manuscript since it will require additional explanation of the connection of the method to 4DVar.

Your item #8 - The issue of observational errors comes up in several comments from both reviewers. Please carefully respond to each of these points by the Reviewers as this is an important aspect of your experiments.

We have responded to each of the reviewers points line by line.

Other critical items raised by the Revs that you have not addressed in your reply include 1) Requests for a more thorough discussion and interpretation of the results (e.g., Rev 2's comment #17, Rev. 1's comment on Page 9) 2) Portions of the text is unclear and needs significant improvements (e.g., Rev. 2's comments #19, 21 3) The relative (pointed out at several places by both Reviewers) vs absolute nature of information pertaining the percentage of variables that need to be observed (e.g., Rev. comments on p.1, p.2, p.3 l.29, p.7 l.10, p.9 l.20, p14 l.8)

Many of these issues have been fixed in the revision. See our response below.

Before I request a second review from the two referees, I respectfully ask that you respond to each and every comment from both reviewers. In your point by point reply, please clearly indicate whether and how you respond to the critical comments and suggestions (i.e., what changes you make in the manuscript). Your replies can be brief but they must clearly indicate your action as to changes you make in the manuscript. If you like, you can copy and paste relevant material from the revised text into your written reply to the Revs' comments. Once we have your full response to the reviews and my comments, I will ask the reviewers for their evaluation of your revised manuscript. I am looking forward to receiving your reply and your revised manuscript.

Respectfully Zoltan Toth

Comment from Referee 1
============================

Techniques for dealing with a sparse observational networks are critically important, particularly for ocean and climate reanalyses that attempt to reconstruct the past state of the Earth system (e.g. Compo et al., 2011; http://www.esrl.noaa.gov/psd/data/gridded/data.20thC_ReanV2.html). The experiment scenarios described here by the authors are perhaps most applicable to the estimation of the global ocean state after the introduction of satellite altimeters, e.g. TOPEX/Poseidon in late 1992 (https://sealevel.jpl.nasa.gov/missions/topex/), with their final set of experiments having a potential application to leverage data from the Global Drifter Program (http://www.aoml.noaa.gov/phod/dac/index.php). Thus from a practical point of view, the time-delay method has potential merit for operational scale data assimilation (DA) and reanalysis.

We agree about the importance of this problem and appreciate your encouraging comments and suggested applications.

Because of such potential, the authors should give a bit more explicit description about how these ideas compare to common methods like 4DVar or the 4D Ensemble Kalman Filter (EnKF), both of which utilize observations over an extended time window. The authors could give a more thorough depiction of how their ideas could be incorporated in these existing systems in order to facilitate a higher likelihood that an operational center might adopt the approach.

We are currently working on connecting the time delay method with other more common methods like ExtKF, 4DVar, and EnKF, which we plan to discuss in a separate paper. Although we considered going into more detail in this paper, we decided it might take away from its main point, which was to demonstrate the benefit of using time delays in a simple geophysical model, and its application to drifter measurements.

That being said, we agree that the core ideas behind these methods have considerable overlap. We have added some discussion on Page 5, lines 25 - 30.

The sea surface height is closely connected to the near surface currents via the geostrophic balance, particularly in mid latitudes. Thus it is expected that unobserved currents would be well constrained by proper estimation of the surface height. For example, sea surface heights and sea surface winds are used to construct an estimate of ocean surface currents for the OSCAR product (http://www.oscar.noaa.gov/index.html).
However, the examples given by the authors could perhaps be described as a supplement for the tropical region where this relationship breaks down. For future work, a natural extension would be to address a slightly more sophisticated example consisting of multiple vertical layers and the modeling of the temperature and salinity components of the density. This experiment would give a better test of estimating

unobserved variables. For example, observing only temperature while estimating salinity is a challenging problem for ocean reanalyses before the Argo era.

Thank you for these suggestions. The complex geophysical processes driving the ocean currents is admittedly not our area of expertise. We will consider testing our methods on such a model as part of our future work.

A brief statement could be made about the applicability of the time-delay approach, for example, to the tropical observing system of moored buoys (TAO/TRITON). These are stationary sensors generating data about once every 10 minutes, but the majority of this data is not used in DA because most global scale ocean assimilation systems use analysis cycles that span multiple days. Even a coupled ocean/atmosphere DA system cycling every 6 hours could benefit from better use of this data. I suggest investigating the TPOS-2020 (Tropical Pacific Observing System) effort for the potential to inform the future development of this and other observing systems (http://tpos2020.org).
A weakness in the chosen experiments scenarios that should be acknowledged is that the approach has not been tested on time-delay observations with errors that are correlated in the time dimension. This is particularly important in ocean DA because errors of representativeness often dominate (versus instrument errors).

Thank you for this suggestion. We were unaware of the TAO/TRITON system and have added a brief statement to our conclusions, with emphasis on future work.

I suggest an experiment, perhaps for future work, in which you run 2 model resolutions. The high resolution run is treated as 'truth', from which observations are drawn. The low resolution model is what you are synchronizing via DA. Set up appropriately, this should give you 'natural' errors of representativeness in the observations that may be correlated in time with the errors of future or past observations. Does the time-delay method still work effectively in this experiment scenario?

This is a good idea. We actually considered including such an experiment, to investigate the impact of finite resolution and model errors arising from subgrid scale processes. Ultimately, we decided to leave these considerations for a future paper, and focus here on the perfect model scenario.

The time-delay method is described in comparison to nudging as a baseline. I would like to see the authors compare a simple 4DVar to the time-delay method as well (via experiment) to give context into how their method compares to a more state-of-the-art DA.

A thorough comparison is planned for a future paper, where we discuss in detail the connection between our method and 4DVar. The current paper now mentions this as future work (p. 6 line 10).

It seems that the time-delay information for the observations and model state applied with what is essentially a diagonal coupling term emulates a similar effect as the cross-covariances that would in effect apply a non-diagonal coupling term to the innovations computed at different times throughout the window. The authors should discuss how the off-diagonal coupling used in most operational DA relates to the diagonal coupling with time-delay observations used here.

We agree with your statements here about the cross-correlations. The off-diagonal terms here arise from the generalized inverse of the time delayed innovations. The diagonal coupling term in time delay space could for instance damp the effect of measurements further in the future, which have more uncertainty due to dynamical instability. A non-diagonal coupling term could also be computed from the observation noise covariance matrix, similar to what is done in 4DVar. When the observation noise is temporally correlated, a time delayed version of this matrix can be used, which has off-diagonal block elements.

While these considerations are important, we have decided to explores them in a future paper that more thoroughly discusses the connection between time delayed nudging and 4DVar. So we have not made any changes to the manuscript to address this question.

The impact of observation error on synchronization via the nudging approach is not addressed very thoroughly. I'd like to see some evaluation of the sensitivity to observation error in the assessment of the method. The authors should describe how their method is impacted by outliers in the observed data. Is the method sensitive to such outliers? I'd like to see an example.

When observation error is present, the model will synchronize to within the noise ball of the `true' solution, when the model is known perfectly and enough observations are present. We recognize however that for many DA methods the goal is to reduce the RMSE below the noise level, but this was not the case here as we chose to consider the sparsity of observations as the dominant effect, rather than observational noise. As a result, we elected to only include a brief investigation, to show that our method is not significantly impacted by very small observational errors.

To be clear though, you are right that enough noise will 'break' this method, or at least severely impede its chances of success. The degree of regularization needed for the generalized inverse of ds/dx is commensurate with the observational errors of the system.

We revised our statements regarding the robustness measurement error to emphasize that these errors are quite small, by meteorological standards. This can be found on Line 22 on Page 6.

"This rescales all data and observed model states to lie in the interval [0,1], so that each state component's contribution to the synchronization error is roughly equal. While this

could make the result sensitive to outliers in the data, it did not appear to be an issue here"

General Technical Corrections:

The manuscript would benefit from more effort in bridging the nonlinear dynamics synchronization terminology with standard data assimilation terminology. This would particularly benefit the readership of this journal.

We agree with this comment and thank you for your detailed suggestions. They were quite helpful, and we have done our best to incorporate them into our revised manuscript.

There is an odd mixing of tenses throughout the citations, and the inconsistency is distracting. The focus should be on the work presented here. Example substitutions are given in the specific comments below. Also, if the author name in the citation is not used as a part of the sentence, please put it in parentheses.

We have revised the paper to follow these guidelines.

The discussion of 4DVar from the conclusion section would be much more useful either in the Introduction or in its own section right after the introduction.

We removed most of the material from the conclusion, opting to save it for a future paper. The remainder has been moved to the end of the Section 2 where the connection with 4DVar is briefly discussed.

The conclusion section is disorganized and needs attention. It should not introduce new concepts in any great detail - instead those should be moved to the body of the paper. Page 14, Lines 24 through the end seem to be a good start to the conclusion section, I advise starting there and reorganizing the rest.

We accept this criticism, and have made considerable revisions to Section 6.

The figures are insufficient. I would like to see a graphical example of the 2-Dimensional domain used for the experiments, including the spatial scale of the layer heights and velocity field. For the drifter experiments, the initial locations for the various experiments should be plotted. Overall, better use of visuals/graphics would greatly enhance the communication.

We have added the two requested plots in Fig. 8 and Fig 10, which were originally withheld due to space considerations.

Specific Technical Corrections:

Page 1:
Abstract:

Line 1:
"The data assimilation process, in which observational data is used to estimate the states and parameters of a dynamical model,"
This is a narrow definition of modern data assimilation. Perhaps reword to indicate that this is one application of data assimilation.

This has been fixed. See lines 12 – 14 on page 1

Line 4:
"Since this problem of insufficient measurements is typical across many fields, including numerical weather prediction,"

It's not clear in what manner the authors feel the measurements are insufficient. A large number of atmospheric measurements are often discarded in operational DA. Is it the quantity or type of observations in the atmosphere that is lacking? The sparsity of observations is certainly a challenge in ocean DA, but this must be considered relative to the timescales and spatial scales of interest. Could the authors please clarify.

We consider the observations "insufficient" if they do not permit synchronization between the model and the data when no prior information is known about the initial state of the system. Our simple synchronization test admittedly only applies to projection measurement operators, though it can be generalized e.g., as in 3DVar. So it is both the quantity and type of measurements that determines whether they are sufficient, as well as the algorithm.

We have added a statement in the abstract and introduction (page 1 lines 3 and 19) to clarify we are only interested here in the spatial resolution of measurements. We also added some further clarification on page 2 lines 8-13 describing which dynamical variables need to be observed (i.e., heights and one of two velocity fields).

Line 5:
"introduced in Rey et al (2014a, b)"
Change to: "introduced [by] Rey et al."

This has been fixed

Line 8:
"For instance, in Whartenby et al (2013) we found that to achieve this goal, standard nudging requires observing approximately 70% of the full set of state variables. Using time delays, this number can be reduced to about 33%, and even further if Lagrangian drifter information is also incorporated."

Change to:
"While Whartenby et al (2013) we found that standard nudging requires observing approximately 70% of the full set of state variables, we find that this number can be reduced to about 33% using time delays, and even further if Lagrangian drifter information is also incorporated."

This has been fixed.

While I understand the comparison to nudging as a baseline/control methodology, could you give an idea of what proportion of state variables must be observed for a more sophisticated DA scheme like 4DVar or an EnKF?

We have not done such an investigation yet with 4DVar or EnKF. Preliminary tests using the ExtKF on the Lorenz 96 model indicate that it can do somewhat better. For that system, standard nudging requires $L_c \sim$ 40-50% of the state variables be observed. The ExtKF can reduce that number to about 20-30%. The tradeoff is that the filter can easily become unstable if not properly initialized. We anticipate $L_c$ for 4DVar and EnKF to be closer to that of the ExtKF, but we will report these results in a subsequent paper.

Introduction:

Line 12:
"The ability to forecast the complex behavior of the earth's coupled ocean, atmosphere system lies at the core of modern numerical weather prediction (NWP) efforts."

This is true, but the current drive amongst many centers is modeling and forecasting the entire integrated Earth System, including for example atmosphere, ocean, sea-ice, land, aerosol, surface waves, ionosphere, etc. I would suggest either saying "the coupled earth system", or further specifying examples beyond simply atmosphere/ocean coupling which has been conducted at some centers for many years.

We agree and make the change on line 12 on page 1

"The ability to forecast the complex behavior of global circulation in the coupled Earth system lies at the core of modern numerical weather prediction (NWP) efforts."

Line 14:
"the observations are completed"
I'm not quite sure what it means to 'complete' an observation. Perhaps this could be reworded.

This wording has been removed.

Line 15:

"The latter is indispensable, even if one has a perfect model, as the accuracy of the prediction is crucially determined by the quality of the estimated initial state values: if the state of the model at the end of observation window is inaccurate, the forecasts will be undependable."

It is my preference to reduce the usage of emphatic adjectives in scientific writings. A possible alternative:

"Even with a perfect model, the accuracy of the prediction is dependent on the quality of the estimated initial state values. Such sensitive dependence on initial conditions was identified by Lorenz (1963; JAS)."

Yes, your suggestion is much more concise. Line 15 – 16 on page 1

"When the model is chaotic, even if it is known precisely, the accuracy of the prediction depends on the accuracy of the initial state estimate. This is due to sensitive dependence to the initial conditions, which was first identified by Lorenz (1963)."

Line 20:
"In our earlier work Whartenby et al (2013) we showed that"
Change to:
"Whartenby et al. (2013) showed that"

Line 8 on page 2: "we refer to the analysis by Whartenby et al (2013),"

Line 23:
I don't think you've specified yet what the 3 dynamical variables are. Even though it is implied by the use of the nonlinear shallow water model, I think you should be explicit.

Line 10 – 11 on page 2

"That is, to achieve accurate forecasts, these methods required measuring the height variable h and at least one of the two velocity variable u, v at each of the N X N grid points"

Page 2:

Line 2:
When the authors mention the use of drifters, does that mean that they are using the position information of the drifters, or simply surface measurements of the model state? Please clarify here.

Position measurements only, but thank you for pointing out drifter height measurements could also be included.

Line 13 -15 on page 11

"Hybrid measurements are incorporated into the time delay nudging method by combining the grid variables and the collective drifter positions

$$\mathbf{R}^{\dagger}(t) = \{[\mathbf{r}^{(1)}(t)]^{\dagger}, [\mathbf{r}^{(2)}(t)]^{\dagger}, \ldots, [\mathbf{r}^{(N_D)}(t)]^{\dagger}\}$$
"

Line 3:

It is not specified whether the spatial distribution of the observations impacts the estimate of L_s. How does this estimate change when all observations are concentrated in one area, or if the observations are arranging in vertical stripes (e.g. like the TAO/TRITON array), or if they are evenly distributed (which can create aliasing artifacts in the analysis) versus randomly distributed.

It does impact the estimate of Ls. Experiments with Lorenz 96 system suggest L_s is higher when observations are concentrated in one area, instead of distributed uniformly. but we have not investigated this in detail. But we have not yet investigated this for the shallow water system.

We intend to address this in a future paper.

There is also ambiguity about whether the state variables are reduced in number only, or in type as well. For example, a collection of global Argo profiles could be cut in half by either (a) reducing the number of floats by half, or (b) eliminating all salinity measurements and keeping only temperature. Both reduce the obs by 50%, but will have drastically different impacts. I'd ask the authors to please be more specific in how the observations are reduced.

Here we consider reduction in the total number of measurements. But you're right, reducing different measurements has drastically different impact. We have not yet come up with a way of quantifying this however. One idea is to look at the change in condition number of dS/dx matrix as various observations are added. More desirable observations should reduce the condition number more than less desirable ones.

We intend to address this in a future paper.

Line 6:
"those at ECMWF Cardinali (2013)"
Change to:
"those at ECMWF (Cardinali, 2013)"

This has been corrected. See Line 5 at page 2

Line 7:
"Evensen (2008); Bennett (1992) should be expected"

Change to:
"(Bennett, 1992; Evensen, 2008) should be expected"

Why is Evensen 2008 ("Data Assimilation, The Ensemble Kalman Filter") cited for 4DVar? I'm sure more appropriate references could be used.
Some references that come to mind, among many others are:
Lewis and Derber, 1985: The use of adjoint equations to solve a variational adjustment problem with advective constraint.
Coutier and Talagrand, 1990: Variational assimilation of meteorological observations with the direct and adjoint shallow water equations.
Zupanski, 1997: A general weak constraint applicable to operational 4DVar data assimilation systems.

Thank you for the advice, we have added the citations by your suggestions. See page 5 line 30.

Line 7:

"While we do not discuss it here in detail, the same results for a 4DVar assimilation method Evensen (2008); Bennett (1992) should be expected."
To which same results are you referring? It is not clear. Do you expect to get all of the same results with 4DVar that you get with nudging? If so, that assertion requires some justification.

Yes, this statement was confusing and has been removed.

Line 11:
"The results appear to be equivalent."
To whom? This statement is ambiguous. Please state clearly whether you found this to be the case in your own research, and if this is part of your research findings, briefly present those results. Otherwise, please list citations of research showing this to be the case. Or, simply remove the statement.

This remark was intended to describe the end result of nudging vs. variational approximation. It has been removed.

Line 13:
"substantial improvement in reducing the number of observations at each measurement time that we report here."
I think you mean "reducing the number of observations needed to attain comparable accuracy in the forecasts or analysis." Please reword.

Line 24 on page 2 has been changed to:

"These outcomes suggest that time delays may be useful for reducing the number of required observations to meet the practical constraints of operational NWP."

Page 3:

Line 1:
Please be careful to acknowledge that this is one application of data assimilation. Model error has many sources. Model parameter estimation can only address a small subset of the many sources of model error.

"In data assimilation we seek to use the information in observations to determine properties of a model that describes the dynamics producing those observations. These properties include unknown parameters as well as the time dependence of unobserved state variables. The model acts as a nonlinear filter coupling the observed states to the parameters and unobserved states."
Perhaps change to:
"We seek to use observations to inform a model of the dynamics producing those observations. Model parameters can be estimated along with the time dependence of unobserved state variables. The data assimilation acts as a nonlinear filter coupling the observed states to the parameters and unobserved states."

Line 8-10 on page 3

"The accuracy of these predictions, when compared with additional measured data in the prediction window t > T, serves as a metric to validate both the model and the assimilation method, through which the unobserved states of the system are determined. This establishes a necessary condition on L that is required to synchronize the model output with the data and thereby obtain accurate estimates for the unobserved states of the system."

Lines 8 and 24:
Equations (1) and (2) on would be more clearly and compactly defined as matrix equations. Particularly in equation (2), the summation notation and the indices are unnecessarily busy.

We have changed our notation to use matrix equations.

Line 9:

"If the dynamics of the system is described by partial differential equations (such as with the fluids in an earth systems model) the ordinary differential equations may be realized by discretizing the partial differential equations on a grid,"
It is worth noting there is a non-trivial amount of discretization error introduced in this process.

We have added this statement on page 3 line 3.

Line 18:

"This is crucial, because it establishes a necessary condition on L that is required to synchronize the model output with the data and thereby obtain accurate estimates for the unobserved states of the system, which are also required to make good predictions."
Change to:
"This establishes a necessary condition on L that is required to synchronize the model output with the data and thereby obtain accurate estimates for the unobserved states of the system."

Line 31 on page 3

Line 27:

"With enough observations L, a sufficiently strong coupling will alter the Jacobian of the dynamical system Eq. (2) so that all its (conditional) Lyapunov exponents are negative."

Please write out the equations describing the Jacobian of the coupled system (2) and the requirements on its eigenvalues to lead to synchronization. I find this is not well documented in the synchronization literature, even in the references provided. The clearest and most detailed description I have found is in the source code written by P. Bryant (http://biocircuits.ucsd.edu/pbryant/#ScientificSoftware). It would be a great contribution of this manuscript to provide a clearly documented reference for this process, particularly for the audience of this journal.

This has been done on page 3 lines 25 - 30.

Line 29:
"This is important, as it establishes a necessary condition on L required to synchronize the model output with the data and thereby obtain an accurate estimate for the unobserved states of the system which are also needed to make good predictions."

It would be preferable if you could state specifically what necessary condition on L is required to synchronize the model. Also, this condition is dependent on the degree of nonlinearity of the system on the timescales of your observation window. A fairly linear dynamical regime would require fewer observations than a more nonlinear regime, measured perhaps using the error doubling time. More detailed discussion would be appreciated.

The necessary condition is that all conditional Lyapunov exponents are negative. We have added additional explanation on lines 25 - 30 of page 3.

Line 17 – 20 on page 3

Line 31:
Change:

"has been shown Abarbanel et al (2009)"
to:
"has been shown [by] Abarbanel et al. (2009)"

It has been corrected. See line 20 on page 3.

Page 4:

Line 3:
"One way to proceed Rey et al (2014a, b); Pazo (2015) involves the recognition that additional information resides in the temporal derivatives of the observations."

This is an interesting idea, that perhaps could be most useful in static observing systems that have high temporal resolution relative to the analysis update cycle. For example, the TAO/TRITON array monitors temperatures at depth at regular 10-minute intervals. However, in NCEP's present operational ocean DA, the data are aggregated to daily or multi-day averages.

Thank you for the suggestion. It would also be interesting to investigate the impact of this averaging on the efficacy of the algorithm.

Line 9:
Since the use of waveform vs. observation time-derivative seems to be the key idea of the time-delay method in this paper, it would be nice if this mentioned up in the abstract and the introduction.

We agree this is a key point but decided to leave this out of the abstract since it would be confusing as time derivatives of observations are not typically used in data assimilation.

Line 10:
Please define $D\_M$ before you use it. Why is the subscript M used? Perhaps something more connected to 'time delays' could be used. Even using 'K' might make more sense since you use an index k for the time delay while you use the index l for the observations of dimension L.

We have kept $D\_M$ so as to follow our previous papers. As an aside, we wanted to use $D\_E$ was changed after a previous referee objected, since it already has meaning in nonlinear dynamics as embedding dimension for time delay reconstruction.

We now define this on page 4 line 10.

Line 12:
I think it should be noted that this requires a static observing system producing observations at all points at regular intervals.

How difficult would it be to extend this approach to the case when the observing system itself is dynamic?

Not difficult, just have to generalize to arbitrary h(x(t),t). The drifters can be considered a form of dynamic observation system.

We did not address this comment in the manuscript.

How does tau relate to delta_t on page 1, line 19?

Added "$\tau$ is the delay, which here is assumed to be an integer multiple of $\Delta$ t." in line 31 on page 4

Line 22:
"DM need only be large enough to effectively increase the amount of information transferred from the L measurements to a value above the critical threshold, Ls."

I understand what is being said here, but it would be nice if the concept of 'amount of information' could be defined more rigorously.

We agree, but are unable at the moment to define this statement more rigorously. Preliminary results with the Lorenz 96 model suggest Ls is related to the average dimension of the unstable subspace, so the embedding only needs to resolve (on average) Ls unstable dimensions. This will be reported elsewhere however.

We could not think of a better way to phrase this statement, so no change was made.

Also, could the authors indicate whether they anticipate D_M to be within the analysis cycle, or perhaps longer than the analysis cycle to the point that observations are 'reused' in consecutive cycles? For example, from page 3, line 6, are the t0, t1, . . . , tN times within the observation window the only candidates for D_M?

Yes, observations are reused over various analysis cycles. So it is a multiple data assimilation scheme. Some clarification was added on page 5 line 57.

Line 24:
The terms "extended space measurement vector" (line 11) and "time delay model vector"
(line 24) seem to refer to the same construct in two different spaces. So it would be helpful to the reader to acknowledge this similarity and use a similar terminology for both.

It has been changed to "time delay measurement vector" in line 13 on page 5.

Line 25:
The notation for the quantity S_i,k looks like a matrix. If the authors would like it to

represent a vector, it should be explained how the elements of the vector are arranged.

The change to matrix notation should make this clear.

Lines 26-30:

You should make the bold notation consistent between here and the equations in (3) and (4). It seems perhaps you didn't mean for (3) and (4) to be bold if the indices indicate a specific element of the matrix.

The notation has been fixed.

Lines 25-31:

I'd like to see these equations represented in their matrix/vector form.

Matrix notation is now used throughout.

Line 31:
Please explain the step between equations (6) and (7) in more detail.

This is now explained in more detail.

Page 5:

Line 1:

"where repeated indices are summed over."
Which repeated indices? Just use the standard summation notation to keep it clear.

Matrix notation makes this clear.

$$\frac{d\mathbf{x}(t)}{dt} = \mathbf{F}(\mathbf{x}(t),t) + \mathbf{G}(t) \cdot [\mathbf{DS}(\mathbf{x}(t))]^{-1} \cdot \mathbf{g}(t) \cdot (\mathbf{Y}(t) - \mathbf{S}(\mathbf{x}(t))), \tag{8}$$

Line 1:

I think you want t to be italicized instead of boldface in G(t).

This  has been fixed.

Line 23:

"So this framework tests the estimation procedure, not the model."
Change to:
"Therefore, this framework tests the estimation procedure absent of model error."

This statement has been removed.

Line 24:
"Removing the issue of model error allows us to assess the weaknesses and strengths of the estimation algorithm and explore in detail the manner in which the unobserved variables are determined."
Change to:
"Removing the issue of model error allows us to assess the manner in which the unobserved variables are determined."

This statement has been removed.

I would argue that the strength of the estimation algorithm cannot truly be assessed until model error is taken into account.

Agreed. However, the algorithm must account for dynamical instability as well, even in a perfect model.

Line 25:
"When successful, it provides confidence that the method may be applied to real data. When it fails, it helps us figure out why."
Change to:
"This is a prerequisite to applying the method to real data."

Change made on Line 3 on page 12

Line 30:
The term "synchronization error" is reasonable when the observations are perfect. However, when the observations are noisy, I believe it's more common in DA to call the term SE the Root Mean Square Deviation (RMSD). If the observations were known perfectly then SE could also be called the Root Mean Square Error (RMSE). I should also point out that in DA the difference [y-x] is often called the 'innovation' within the context of the update procedure, and the "OMF" (for observed-minus-forecast) outside of that context.

In general, it is my preference to avoid the term "error" (e.g. SE or RMSE) when the observations are not known perfectly (i.e. in real-world experiments) but instead use RMSE. The word 'error' tends to give the wrong impression that a smaller value is better, which is not necessarily true when the observations are noisy.

This is a good point, which we had not previously considered. We have modified section 3 to use RMSD.

Page 6:

Line 3:

"so that each state component's contribution to the synchronization error is weighted approximately equally."

It seems that the scaling applied to the observed variables could potentially be highly susceptible to noise or outliers. Is this the case?

You are correct. This works best when the noise is relatively small. We have amended our statements on Line 25 on page 6.

Line 9:
"It is crucial to compare SE(t) for both estimates and predictions"
The DA terminology for that would be 'analyses' and 'forecasts', respectively.

We have made an effort to incorporate more DA terminology in our revisions.

Line 10:
"It is crucial to compare SE(t) for both estimates and predictions, as the former is just a 'fit' involving measured quantities, while the latter relies on accurate determination of the unmeasured variables as well."

I think it is worth mentioning again that the former is called "observation-minus-analysis (OMA)" and the latter is called "observation-minus-forecast (OMF)" in operational applications.

Yes, we now refer to them as RMSD.

It is well understood in the atmospheric DA community that the OMFs are the more important measure for improving forecast skill, as OMAs can be set arbitrarily small by design.

Yes, we apologize for being a bit redundant about this. Outside of DA, many people just use estimates or "fits", so this has become an important point for us. We have revised these statements based on your suggestions.

Lines 12-14:
"Accurate estimates alone are not sufficient to validate the model or indicate the success of the estimation procedure, as they do not [contain] any information about the unobserved states."

They do contain information about the unobserved states from past observations propagated through the model from previous analysis cycles. The more important distinction is that the OMAs can be made arbitrarily small, but the observations contain errors and so this is not an indication of a better estimate. Rather, we would like to see consistent statistics of the OMFs over many cases showing a reduction in mean departure (assuming there are no biases in the observation errors).

We have removed this comment. Our point was that, the estimation error (or OMA) at the end of the estimation window does not provide any information about whether the unobserved states are correct.

Line 15:
"We have previously shown that when the synchronization error Eq. (9) decreases in time to very small values, the full state"

I don't think this can be true if there are large/outlier observation errors. At that point you would be fitting the noise and likely disrupting the state estimate and subsequent forecast. Do you mean the average SE decreases in time?

Agreed. This statement has been removed

Section 4, Nonlinear Shallow Water Equations:

Line 26:
"We argue that the results presented here, for this simplified model, will be applicable for establishing the initial state of those models and predicting their subsequent behavior."

There is insufficient justification for that argument presented here. There are some additional experiments that must be done with this method before this claim can be stated confidently. A next step would be to apply the time-delay method an example case consisting of multiple layers while computing the temperature and salinity components of the density. This experiment would give a better test of estimating unobserved variables, though this is probably beyond the scope of this work.

We have modified the statement on page 7 line 10.

Page 7:
Line 5:
I'm curious what impact the wind forcing had on the results. Have the authors tried experiments with multiple wind fields?

No, we have not.

Line 7:
"With these fixed parameters the shallow water flow is chaotic, and the largest Lyapunov exponent for this flow is max = 0.0325/h 1/31 h."

Please explain how the Lyapunov exponent was computed for this case.

Line 18 – 20 Page 7

"With these fixed parameters the shallow water flow is chaotic, and the largest Lyapunov exponent for this flow is estimated to be  max = 0.0325/h =1/31h, measuring the average growth rate of random perturbations"

Line 9:
You should clarify that the grid size {16,32,64} is changing resolution over the same domain boundaries.

We have added this. See page 7 line 25.

This result is interesting, because it implies that given an observing network, you can determine the exact scale of features that can lead to synchronization by using that observing network. There is significant interest in transitioning climate models to higher resolutions going into the future (e.g. http://cpo.noaa.gov/ClimatePrograms/ModelingAnalysisPredictionsandProjections/OutreachPublications/MeetingsWorkshops/while at the same time a number of new satellites are coming on with much higher spatial resolution (e.g. SWOT Altimetry http://journals.ametsoc.org/doi/abs/10.1175/JTECH-D-13-00109.1). The authors are encouraged to explore this concept in more depth.

Thank you for the encouragement.

Line 10:
"we estimated that approximately 70% of the D = 3N^2 degrees of freedom must be observed in order to synchronize the model output with the data"

What observing network did you use? I.e. what was the distribution of observations? Were they stationary? Are they observed all at once, or throughout the observation window. These details will impact the number of observations needed.

Additional details have been added on page 7 line 27.

Line 16:
"We are confident that despite the numerical challenges associated with scaling the algorithm up to larger D, the results presented here for N = 16 will also remain valid for higher grid resolution."

Does this also hold if the grid resolution is kept constant but N is increased by increasing the domain size? 50km grid spacing at the equator is about the resolution of many global ocean models run in operations. Do you expect a ½ global model to require around 70% coverage? A dynamic observing system tends to require fewer observations to maintain synchronization (e.g. see Penny, 2014, where the Lorenz-96 system is constrained by randomly located observations http://journals.ametsoc.org/doi/abs/10.1175/MWR-D-13-00131.1). A methodology for calcuating Ls for more realistic observing systems would be a valuable contribution.

This statement has been removed, and the issue will be considered for future work.

Line 18:
"In the discussion above, which included reference to the lectures of [Cardinali] Cardinali (2013), we see that the requirement of having to observe[d] 70% of the model dynamical variables exceeds the measurements now available by at least a factor of two; more if the NWP model is larger [yet]."
Change to:
"In the discussion above, which included reference to the lectures of Cardinali (2013), we see that the requirement of having to observe 70% of the model dynamical variables exceeds the measurements now available by at least a factor of two; more if the NWP model is larger."

This change has been made.

Page 8:
Section 5, Results with Time Delay Nudging for the Shallow Water Equations:
The equations on this page are missing equation numbers.

Equation numbers are not used for equations that are not referenced.

Line 9:
These look a bit like a stream function but usually that's written in the other form. Just make sure this is what you intended.

It has been corrected. See Eqn 11 on page 8

$$u^{(i,j)}(t_0) = A_0 \frac{\partial \psi(\mathbf{r}^{(i,j)})}{\partial y} \qquad v^{(i,j)}(t_0) = -A_0 \frac{\partial \psi(\mathbf{r}^{(i,j)})}{\partial x}$$

Line 16:
Using a diagonal coupling matrix G ignores spatiotemporal correlations between points. Do you have any comments about the implications of using a diagonal G?

None, other than it's the simplest choice we could make. As discussed earlier, if the observation noise has temporal correlations we could use its covariance matrix to determine a non-diagonal G, similar to what is done in 4DVar. This will be reported in a subsequent paper.

Lines 16-19:
I thought G was the time delay space coupling and g the physical space. But, here is says g is the time delay space coupling term. Is this a typo?

Yes, it is a typo and has been corrected.

Line 17:

"These are chosen because the height values are several orders of magnitude larger than the flow velocities."

You could consider normalizing the innovations by the corresponding observation errors (i.e. standard deviation used for Gaussian noise).

This poses a problem when the observational errors are small.

Line 23:
Please cite a reference for calculating the average mutual information, using whatever method was used here.

The citation has been added.

Line 24:

"Furthermore, the results were reasonably stable to changing its value within a few delta t."

I'd like to see some analysis of the sensitivity to delta t to justify this statement.

The choice of delta t is based on the paper by Sadourny (1975). We do not intend to dive into the stability of the time discretization.

Lines 26-27:
Do you use a time-delay extending before the beginning of the analysis cycle window [0, T]?

The time–delay analysis ranges from the analysis cycle window [0,T] and it may extend beyond the window when constructing the time-delay vector near the end of the window T. We have added some clarification on page 5 line 25.

Page 9:
Line 3:
"Since DM = 8 produces error values several orders of magnitude smaller than those obtained with DM = 6, we expect the state estimates x(T) obtained with DM 8 to be quite accurate when compared with the estimates for DM = 6."

There seems to be a critical point between D_M=6 and D_M=8, what are the results for D_M=7? How do you explain this bifurcation? Is there a waveform that can only be resolved at this time delay length?

We are unable to explain this result at this time. For the Lorenz 96 model, it is related to the average dimension of unstable subspace. We have not looked into the length of the waveform however.

When D_M = 7, synchronization depends on the choice of initial condition, so we decided present the cases for D_M = 6 and D_M = 8 which do not depend on the initial condition.

Line 11:
"Just a reminder note here, we used L = 256 = 33% of the total 768 dynamical variables as observed, then used time delay information on the waveform of the measurements to provide the required additional information."

This reminder is not necessary here in the results section. Instead, the clarification should be made earlier, perhaps when the value of Ls is stated on page 7 line 15.

These statements have been removed.

Line 20:
"As this point is the key theme of this paper, we take the liberty of repeating that the number of physical measurements is just 33% of the overall dynamical variables."

Yes, but the height has a strong relationship with the currents. What kind of results do you get if you observe only the u or v components of the velocity instead of height? Is 33% still sufficient?

It will undoubtedly change. We focused on heights since we considered them easiest to measure in practice using satellites.

Line 26:
"in accordance with our previous results in Whartenby et al (2013)"
Change to:
"in accordance with Whartenby et al. (2013)"

This has been changed.

Lines 31-32:
"When enough information is available, and the coupling is strong enough, these conditional Lyapunov exponents will all be negative, allowing the coupled systems of data and model output to synchronize."
Do you have a means of computing the conditional Lyapunov spectrum for this system? If so I would like to see these results presented.

We do not have the means to do this at the moment for this model.

Page 10:

Line 7:
"in a true experiment, the success of the assimilation procedure must be evaluated against the predictions - not the estimates."

Again, if you are going to discuss real-world problems, I suggest using the appropriate terminology: "forecasts - not the analyses."

This has been removed.

Line 19:
"We remark, however, that the overall space of parameters appearing in our formalism has not been thoroughly explored and that by further adjusting these parameters"

One obvious example to maintain prediction skill with reduced observations would be to use non-diagonal coupling terms in your nudging (as is standard for operational data assimilation).

Yes, the number of required observations can be further reduced using an `optimal' coupling, computed for instance using the Riccati equation to approximate the error covariance. We will save this investigation for future work.

Line 22:
"One would expect, that at some point the resolution should be high enough to not necessitate further measurements."

I don't understand this statement. In a realistic system, as the resolution increases, the resolved features also increase. In that case, I would expect the required observations to increase super-linearly. I suggest more experimentation should be done before making such a claim.

In practice, yes. For this simple model however, there may be a resolution at which it is fully resolved. We have revised this statement on page 8 line 3.

Section 5.1 Measurements with Gaussian Noise

Line 24:
"In operational data assimilation in meteorology, one challenge is that the measurement contains observation error."

This is true in all data assimilation, regardless of the domain.

Change to:
"In operational data assimilation, observations contain measurement errors and systematic biases."

This has been removed.

Line 29 and 30:
Missing equation numbers

Equation references are not needed here

Line 31:
"and we selected C_data = 106 and C_height = 1652."
Please list the scale of the observation errors used in the experiments shown in figure 7 within the text.

The expression has been changed as follows (see Line 29 on page 10)

"We now repeat the above calculations for L = 252 with Gaussian noise N (0, σ) added to the height observations. A comparison is shown in Figure 7 for σ  = { 0.2,0.5 } and D_M = {8,10}"

Page 11:

Line 2-3:
"The time delay nudging method remains robust under imperfect observations."

I would anticipate there are observation errors that can cause outliers large enough to 'break' this method. Have the authors found such cases?

Yes. We have changed this to say it remains relatively robust to small observational errors.

Section 6. Using Drifter Data with Time Delays

Line 9:
The authors should mention here if they append the drifter coordinates to the state vectors, or if they convert the drifter positions to an Eulerian velocity measurement.

We append drifter positions to the state vector. See on page 11

$$\mathbf{Y}_{drifter}^{\dagger}(t) = \{\mathbf{R}_{data}^{\dagger}(t), \mathbf{R}_{data}^{\dagger}(t+\tau), \dots, \mathbf{R}_{data}^{\dagger}(t+\tau(D_M-1))\}$$
$$\mathbf{S}_{drifter}^{\dagger}(t) = \{\mathbf{R}_{model}^{\dagger}(t), \mathbf{R}_{model}^{\dagger}(t+\tau), \dots, \mathbf{R}_{model}^{\dagger}(t+\tau(D_M-1))\}.$$

Lines 16-19:

"After the initial deployment, the drifters move between grid points providing information not available from grid point measurements alone."

In order for this to be true, the drifter positions shouldn't be determined solely by the grid points. For example, they might be computed on a finer model grid. However, in the next paragraph it is said:

"The dynamics of drifters is described as two-dimensional fluid parcel motion on the

surface of the water layer. Since the positions of drifters are continuous values, the velocities of the drifters are estimated by a smooth linear interpolation"

So, if the velocities are determined by a smooth linear interpolation, then aren't they determined from the grid point measurements alone?

In our simple model, yes, since their positions needed to be simulated. In reality, no. This statement has been removed nonetheless.

Line 28 and 30:
Missing equation numbers

Equation references are not needed here

Line 28 looks as if it should have a left bracket at the beginning of the line.

It has been fixed.

Page 12:

Perhaps you might also consider an additional case with L = 208 + 20 while N_D = 0 to ensure a fair comparison, or at least give more perspective on the impact of having a portion of the observation drifting versus stationary.

When N_D = 0 and L = 208 +20, it's similar to the case in the last subsection. Moreover, the data from heights and the data from the drifters are of different types.

Lines 5-6:
"We have further investigated how the geographic distribution of the drifters influences the size of the synchronization error, although these results are not displayed here."

Why not? These results would be interesting.

Due to the scope and length of this paper, we hope to include this result in the future work.

Line 11:
Is it possible to generate an estimated current velocity based on the wind field and heights? Does that velocity correspond well to the drifter observations?

We are not sure how to do this. Can you please explain?

Section 7. Conclusion

Line 15:
"In an earlier paper Whartenby et al (2013) we showed that using standard nudging"

Change to:
"Whartenby et al. (2013) showed that using standard nudging"

This has been changed

Line 22:
"realistic and complex models of the ocean, atmosphere system"
Change to:
"realistic and complex models of the ocean [or] atmosphere"

This has been changed

Line 26:
This discussion about 4DVar should be a section either in the introduction or just after, giving the context of the time delay nudging method relative to the 4DVar method.

This discussion has been removed.

Page 13:
Most of the discussion on model error should be moved to the body of the manuscript. It does not seem appropriate in the conclusions.

It has been removed

Page 14:
Line 8:
"The framework presented here allows one to directly estimate the minimum number of observations at each measurement time required for accurate predictions, Ls"

I'm not sure I would characterize it as a direct method for estimating $L_s$. A brute force application of many values until achieving synchronization seems more indirect. A more direct method would be desirable.

Agreed, we are working on a more direct method. This statement has been removed.

Line 12:
Because 'almost surely' has a very specific mathematical meaning, I would suggest to change:
"real data will almost surely fail"
Change to:
"real data will likely fail"

Agreed. See line 27 on page 12.

Line 13:
"On the other hand, when the process succeeds, it increases confidence that predictive

failures associated with the assimilation of real data arise from inadequacies in the model." Or, inadequacies of the observing system.

This has been removed.

Line 14:
"When the model is wrong, as it typically will be in practice"
Change to:
"When the model is imperfect, as will be the case in practice"

This has been rephrased. See p 12 line 23.

Line 16:
"In other words, when predictions fail, our strategy provides a useful diagnostic framework to help determine where to concentrate our efforts: improving the model or collecting more data."

This statement should be tempered or removed. I don't see such a tradeoff being made in a real-world scenario, in short because there are far more considerations that go into such decisions than whether one can get a better analysis at a given time. For better or worse, these efforts tend to be independent. There are obvious reasons to develop both the model and increase data collection in parallel. For example, model development can always be postponed, but data collection can never be repeated.

Valid point. However, it is helpful to know what the limiting factor is. This statement has been removed. See p 12 line 23.

Lines 18-19:
"The inclusion of time delays comes of course with an additional computational cost, mainly associated with the integration steps required to construct the time delay vectors and its Jacobian, as well as solving for the perturbation itself."

This should be stated at the very beginning when the method is first introduced.

Agreed. This sentence has been removed.

Line 24:
This should be the leading paragraph in the conclusion section.

This has been removed.

Page 19:
Figure 2:
This is not a complete sentence: "In accordance with the synchronization error results."

This has been fixed

Comment from Referee 2

============================

Abstract. The authors should not introduce in the abstract alone important quantities, such as the critical threshold, Ls. This must be re-introduced in the Introduction or at least somewhere else in the main body of the article. Note for instance that it is only in the abstract that it is stated clearly that the work deals with chaotic dynamics. While this is patent in the rest of the paper it must be said explicitly.

It has been removed from the abstract and defined on page 2 line 12.

2. Introduction. Lines 15-17, page 1. The statement is strictly true only for chaotic systems (this relates to my previous comment).

This statement has been rephrased.

"When the model is chaotic, even if it is known precisely, the accuracy of the prediction depends on the accuracy of the initial state estimate. This is due to sensitive dependence to the initial conditions, which was first identified by Lorenz (1963)."

3. Page 2, lines 7-11. When talking about the necessity to control the unstable modes, it is relevant to mention methods, like the Assimilation in the Unstable Subspace, in which the analysis update is explicitly designed for this purposes; (see e.g. Palatella, L., A., Carrassi and A. Trevisan, 2013. Lyapunov vectors and Assimilation in the Unstable Subspace: theory and applications. J. Phys. A: Math. Theor. 46, 254020)

Agree, the citations have been added. See line 3 – 5 on page 6

4. Page 2, line 12. The reference is Pazo et al., 2016 (Pazo, D., A. Carrassi and JM. Lopez,
2016. Data Assimilation by delay-coordinate nudging. Q. J. Roy. Meteor. Soc. 142, 12901299)

Yes, this typo has been corrected. See Line 8-9 on page 15

5. Page 2, line 16. Ls is not properly defined. The authors should explain a bit more precisely what it is meant by "critical threshold"?

The definition has been further clarified on page 2.

6. Page 3, line 5. Notation or text must be improved. Is y a L-dimension vector? If so, you better state that you take L observations that are then collected in the observation vector y.

Yes y is an L dimensional vector, which is constructed by the measurement from L grids. The switch to matrix notation should help.

7. Page 3, line 13. Again notation or text must be improved. The equation $y_l(t) = x_l(t) +$ noise suggests that L = D which is not the case in your experiments, and the fact that $L_D$ is indeed one of your key point. You should say that you use an operator H that only observes a portion of the state-vector (mainly the heights in the experiments that follow).

We have introduced the projection operator H.

8. Page 3, Eq.(2) and lines 24-30. Eq.(2) requires observations at each time-step of integration, something which is usually obtained by interpolating in between successive observations in real applications. For the sake of completeness, it must be also added that the condition of a negative conditional Lyapunov exponent implies the setting for the strength of the nudging forcing, g and not just on L. In classic literature in fact this is usually the case, and the observational network is given.

Agreed. See Line 20-25 on page 3

"With enough observations L, and a sufficiently strong coupling G(t), this control term alters the Jacobian of the dynamical system Eq. (2) so that all its (conditional) Lyapunov exponents are negative — see e.g., Pecora & Carroll (1990); Abarbanel (1996); Kantz & Schreiber (2004)."

9. Page 4, Eq.(7). I think g and G should not be bold.

Now, the notation has been unified. Line 11 on page 3.

$$\frac{d\mathbf{x}(t)}{dt} = \mathbf{F}(\mathbf{x}(t), t) + \mathbf{H}^\dagger \cdot \mathbf{G}(t) \cdot \left( \mathbf{y}(t) - \mathbf{H} \cdot \mathbf{x}(t) \right).$$

10. Page 7, line 15. How is Ls obtained? Does it come from the simple nudging case Eq.(2)?

The value of Ls given by Whartenby et al (2013) was computed from standard nudging. This has been further clarified on page 2.

11. Page 7, line 18. "Cardinali" is written twice. Line 19:" observed" should be" observe".

The typos have been corrected.

12. Page 8, line 20. The sentence about parameter estimation relates to the chosen" perfect model" scenario. It would be better if the author states this clearly.

This statement has been removed.

13. Page 8, line 22-24. You might want to say something more on this regard. How is it optimized? What does it mean average mutual information? Is it the time decorrelation scale?

Average mutual information is a heuristic technique for estimating an appropriate tau. See e.g., Abarbanel 1996.

14. Page 9, line 4-6. Are you computing the error using only the observed components of the state vector also for t > T?

Yes, this has been clarified by changes to Eq. 9.

15. Page 10, line 1. Do you mean Fig.5?

Yes, the typo has been corrected.

16. Page 10, line 5-6. Do you have suggestions on how to select the coupling?

A suitable choice of g must make the conditional Lyapunov exponents negative. If the observation noise is known, it can be incorporated as well like what is done in 3DVar.

17. Page 10, line 11-13. In fact, the values chosen for L are very close to each other and results highlight a strong sensitivity of your method to this (error diverge when L = 248). Can you comment more on this? Another aspect regards how those observations are placed. One can always achieve a better control of the error by a proper deployment of the observations (possibly with the use of target observations). What would it happen if observations were denser in the proximity of the most dynamically active areas?

This is no different from the typical divergence with simple nudging methods and L < L_s, and different observation schemes will give a different values of L_s. We have not looked into any different observation schemes for this paper.

18. Page 10, line 18-22. The final paragraph of Section 5 is important but it necessitates some improvements: (1) You might want to say that, as known from classical synchro-nization results, the optimal forcing strength (g and G in the present context) depends on the number and distribution of the observations; (2) the conclusion relating the model resolution and observations network is unclear. Even with a high resolution model one may still necessitate a growing number of observations to keep under control the unstable modes. Please clarify this point.

We have revised these statements, they should be more clear now.

19. Section 5.1. A number of details are unclear in the present version of the Section. In particular one gets easily confused by the mix of information on how observations are

made noisy given in the caption of Fig.7 and the fields" data" in lines 29-30. (1) What is φ in the first equation? Did you mean ψ?; (2) From the 2nd equation one sees that the observations have zero mean, which seems inconsistent. (3) In the experiments so far you have only observed h, why are you then showing how observational noise is simulated for the velocities if the latter are not assimilated? (4) What are the values of C height and C data and how are they chosen? We do not know how these values scales with respect, for instance, to the system climate variance in the same variables. It is consequently impossible to judge to which extent this observational error is small or big. (5) Please make consistent the text in Section 5.1 with the Fig. 7 caption. From the latter one learns how observations are being perturbed with normal distribution with variance σ.

1. Yes. This was a typo.
2. Also a typo.
3. I do not understand this question.
4. This has been revised to be more clear.
5. This has been done.

20. Page 11, line 6. The reference should be Mariano et al (2002). Isn't?

Yes. It has been corrected.

21. Page 11, line 18-30. This part is very interesting but key details are missing on the state-augmentation formulation that the authors seem to have used to incorporate drifts. Please improve presentation on this aspect and provide more details.

Additional details have been added.

22. Page 13, line 23-25. Although its meaning may be clear to a reader from the data assimilation community, Rm is undefined? It may be convenient to define it in relation with Eq. (12), i.e. with Rf.

This paragraph has been removed.

23. Page 12 - 13. The long discussion on the equivalence with smoother (4DVar) is very interesting but, in my opinion, very badly placed in the middle of the conclusion. That is not a conclusion, but rather a discussion. I think the authors should move it in the main body of the manuscript, perhaps when the method is presented and before the numerical results. In any case an independent section would be ideal.

24. Page 14, line 27-34. This comes too late and it would be better seen at the beginning of the Conclusion, when you recall the motivation behind the research effort.

This has been removed.

25. Page 15, line 1. You have repeated already many times the model you have adopted.

This has been removed.

26. Page 15, line 5-6 and line 13. Check the reference to Cardinali (2013).

This has been corrected.

27. References. Some entries in the list have typos that require corrections, particularly in the page information which has "?" instead of "-". This is the case, for instance, of Kuznetsov et al, 2003, but the problem is present elsewhere as well. Make consistent use of journal names abbreviations throughout all entries.

The citations have been corrected.

28. Figure 6. Correct labels and box in the top panels.
The labels have been corrected.

---

## Author Response (AR3)

We would like to thank both reviewers for their careful reading and thoughtful comments. Our responses are given inline as follows.

The authors have greatly improved the clarity, conciseness, and focus of the manuscript. It reads well and should be considered for publication with minor revisions at detailed below.

As a general technical comment, try to avoid 2-sentence paragraphs.

I think it would be a useful addition to the presentation to have the first figure of the manuscript as an example representation of the model domain, similar to Figure 10, but just showing a snapshot of an example flow and surface height field that is representative of the model's behavior under the given surface forcing.
We have added three representative plots: one at the initial time, one after 30m, and one after 30h. (Fig.1)

p.2, line 3:
I suggest to change the citation format from " (Abarbanel (2013))." to "(Abarbanel 2013)." for this and all following similarly-formatted citations for conciseness.
Thank you. We were not aware this was an option.

p.2, line 8:
This is a 2-sentence paragraph, the first of which is a run-on sentence. I suggest breaking the first sentence up to be more concise.
p.2, line 10:
Remove "In other words," since what follows is adding information rather than rephrasing the previous statement.
Paragraph now reads:
To clarify these ideas we refer to the observability study given by~\cite{mwr13}, which evaluated the performance of familiar nudging methods on chaotic, shallow water flow. The flow was simulated on a $\beta$-plane defined by a square grid with uniform spacing $N_\Delta$, periodic boundary conditions, and driven by Ekman pumping. Poor predictions were obtained unless the height variable $h$ and at least one of the two velocity variables $u$, $v$ at each of the $N_\Delta \times N_\Delta$ grid points were measured. In other words, accurate forecasts required direct observation of roughly $70\%$ of the $3N_\Delta^2$ dynamical variables.

p.2, line 23:

"These outcomes suggest that time delays may be useful for reducing the number of required observations to meet the practical constraints of operational NWP"

I suppose there is some degree of interpretation here as to what qualifies as a 'countable' observation. I would call a single instrument making 10 consecutive measurements at the same point 10 observations.

In that sense, the use of time-delay methods could make better use of a fixed number of observations that might currently be ignored.

What needs clarification is whether L refers to the number of distinct observation locations, or the total number of distinct measurements in a time window? Please make sure this is clear.

The former is correct, except that it is the number of observation locations times the number of observations made at that location, since for instance h,u,v are all measured at the same gridpoint. We have clarified this in the second paragraph of p1:

Here we consider an idealized situation where a perfect dynamical model describes the deterministic time evolution of a set of $D$ state variables. We assume that $L$ measurements are made at uniform time intervals $\Delta t$ within an observation window of length $T$, so the total number of distinct measurements is $L\times(T/\Delta t + 1)$.

Also, to the second paragraph of section 2.

The total number of measurements in the observation window is $L\times(N+1)$.

p. 2, line 27:
Connect line 27 into the paragraph on the next line.

Done.

p.3, line 25:
Is capital Phi the state transition matrix in this context? Could you state its role in this case?

Yes, it is the linearized state transition matrix. We have modified the sentence to read:

where $\vec{\Phi}(t,t') = \partial\vec{x}(t)/\partial\vec{x}(t')$ is the linearized state transition matrix. Its time evolution is described by the variational equation

p. 4, line 29:
eq 4: Do you intend for tau to be negative so that adding increasing multiples of tau goes backwards in time? It may be more clear to assume tau is positive and subtract multiples of tau to illustrate the delay aspect of the approach. If not, you may want to make it clear immediately that your time 'delays' are actually forward in time given your indexing framework.

Yes, the 'delays' here are forward in time. We have modified this paragraph as follows:

...and $\tau$ is the delay, which here is assumed to be a positive integer multiple of $\Delta t$. Also, note that the term `delay' here is not used in its usual sense. Rather, this method involves an advanced formulation, which for positive $\tau$ incorporates observations at later times. Both formulations are acceptable however.

We also modified the discussion of the connections with 4DVar on p5-6.

p.4, eqns 4, 5:
Thanks, this is much clearer than the previous presentation.

p. 5, line 20:

It may be helpful to clarify the definition of 'delay' when the delay is first mentioned.

Agreed, see response above. P.4 line 29

p. 6, line 2:

Change "nudging [using] truncated" to "nudging [uses] truncated"

Fixed.

p.6, line 14:

This isn't a complete sentence: "Namely, the shallow water equations."

Fixed.

Instead, we focus its application to a core geophysical model: the shallow water equations.

p.6, line 18:

"data is taken" to "data are taken"

Fixed.

p. 7, line 24:

"by measuring the average growth rate of random perturbations."

I think a little more detail is needed here. Are you using a 'bred vector' type method? If so, the scales captured are determined by the magnitude of the random perturbations, the rescaling interval, and the norm used. These parameters impact the scales of the instabilities that are amplified and identified by the algorithm. Please provide more detail, as it is difficult to interpret this description.

For bred vectors, e.g., see Toth and Kalnay (1997)
http://journals.ametsoc.org/doi/full/10.1175/1520-
0493(1997)125%3C3297%3AEFANAT%3E2.0.CO%3B2

The details of this calculation are given in our Whartenby 2013 paper.

> *... we selected two sets of very close initial conditions and integrated the two solutions of the equations forward in blocks of 4000 time steps of 0.01 h each. During each block of 40 h, we tracked the RMS distance between the two slightly perturbed solutions to the shallow-water equations and evaluated the largest Lyapunov exponents (LE) by approximating the growth of this error as e LE3t . Following the standard procedure in studies of nonlinear dynamics, when this error grew large, we rescaled the values of the two orbits to have a very small value of their distance and then tracked growth of the distance between the same two orbits for another block of 40 h in time. We did this for 80 blocks of time.*

A reference has been added in the text.

With these fixed parameters the shallow water flow is chaotic, and the largest Lyapunov exponent for this flow is estimated to be $\lambda_{max} = 0.0325/h \approx 1/31h$ by measuring the average growth rate of random perturbations. The details of this calculation are given by \citet{mwr}.

p. 8, table 1:

Could you clarify - is the spacing delta X and delta Y consistent across all resolution experiments {16,32,64}, or is the total domain size constant?

Total domain size is constant. Added clarification in p4 of section 4.

The total domain size is constant $800 \times 800~\text{km}$ and periodic boundary conditions are enforced.

p. 8, table 1:

The authors list the Coriolis parameter f0 and the Rossby parameter (meridional derivative). Please also list the corresponding latitude for this f value (e.g. at the center of the model domain).

Central latitude of the beta plane ~3.6 deg has been added to the table

p.9, lines 18-24:

Please mention that the results with D_M=7 reached high accuracy in some cases, dependent on initial conditions. These 'boundary' cases are interesting to identify for future study.

Added the following:

In addition, with the choice $D_M=7$ some initial conditions synchronized while others did not. Further analysis of this `boundary' case is an interesting area for future study.

p. 10, line 30:

Change "The same [striking] improvement" to "The same improvement"'

The use of extreme adjectives of this type does not enhance the presentation. The results speak for themselves and the readers can determine whether or not they find them 'striking'.

We agree, the adjective has been removed.

p. 10, line 25:

I'd like to see one case additional where the noise is large enough to break the synchronization.

Added the following:

Furthermore, results fail to synchronize when the magnitude of the noise gets large enough. This transition occurs at roughly $\sigma = \{1.3,2.0\}$ for $D_M = \{8,10\}$ respectively.

p. 11, line 10:

"The dynamics of drifters are described as two-dimensional fluid parcel motion on the surface of the water layer"

Technically, wouldn't the drifters modeled in these equations be representative of the vertically integrated layer velocity? This may be a better interpretation anyway, since in reality drifters are usually representative of a given depth. For example, the GDP drifters are drogued at 15m. Surface floating drifters may be adversely impacted by winds and thus not accurately represent the near surface currents.

We have changed our statement to read

...as two-dimensional fluid parcel motion near the surface of the water layer

p. 11, line 14:
"Hybrid measurements are incorporated into the time delay nudging method by combining the grid variables and the collective drifter positions"
Is it the drifter position you are appending, or is it really the drifter id and dimension label as a type of coordinate space for the drifter data? The position data should be analogous to the velocity and height measurements, while drifter id's are analogous to the model grid points.
The 2D drifter positions are appended to the state be vector. They provide analogous velocity and height information in between gridpoints. But that information is not used directly, say by approximating the drifter velocity and interpolating it to the nearest gridpoint. Rather the drifter information is transferred to the estimate through its dynamical model dr/dt = {u(t),v(t)}. For the simulations, the u,v fields had to be interpolated from the true solution at the drifter positions. Whereas in reality, this data will just be provided.

Does this answer your question?

p. 11, line 16:
As a warning, the boldface capital R is typically used in DA to represent the observation error covariance matrix, and so the choice to use it in this context may cause unnecessary confusion.
Thanks for pointing this out. It has been changed to \xi.

p. 12, line 4-9:
I'd like to see one more case where the approach to forming the initial conditions matching the previous cases is used, and sufficient drifters are added to achieve synchronization. This would make it easier to compare the findings in this section with results from earlier sections.
We ran these cases, but the results are inconsistent and depend on where the drifters are initialized. As such we have decided to leave these results out of the paper. Obviously there is more work to be done here.  However, we have modified two paragraphs in this section to make this more clear.
We consider cases with $N_D = 0$ (no drifters), as well as $N_D = 20$ and $N_D = 64$, in addition to $L = 208$ and $L = 128$ grid observations. Example plots showing the initial locations for $N_D = 20$ and $N_D=64$ are given in Figure 8. While many runs were successful using the same setup described above, the results were somewhat dependent on where the drifters were initialized. These results are not shown.
The consistency of the results improved by choosing the initial estimate to only in magnitude from the true solution, rather than in both both phase and frequency as was done above. Specifically, for the results reported below the initial conditions of the data ...

Page 12, line 21:
Please change:
"when the model is [wrong]" to "when the model is [imperfect]"
If the model is 'wrong' in practice then it cannot not be used for forecasting or data assimilation. It must

be reasonably accurate to produce any reliable forecast skill.

Fixed

page 12, line 22:

"this methodology provides some idea as to whether the model is at fault, or whether more observations are needed."

I suppose I need more explanation for how the methodology provides this information. I understand that it indicates for a given model you can identify how many observations should be sufficient for synchronization. So are the authors extrapolating that then if the model does not synchronize when using real data then the model is to blame? There are subtleties in this process that make this statement seem like an oversimplification, particularly regarding the assimilation of satellite data.

The point we are trying to make is that, if synchronization occurs with simulated data but not with real data, the model may be the limiting factor. Whereas if it doesn't succeed with simulated data, given the constraints on observability, then the observations are likely the limiting factor. However our argument is muddled by the fact that (as you mention) the synchronization methods we've described here don't apply to situations where the measurement operator is nonlinear, such as for assimilation of satellite radiance data. We have changed this argument to read:

Since the successful analysis of simulated data a prerequisite for success with real data, this methodology provides a way to assess where one stands with respect to critical observability limits of the system at hand.

Figures 4,5,6:

It may just be my pdf rendering, but I see two plots on the top row but only one plot in the center of the bottom row. It appears that one is missing. Please double check to make sure there are reproduced as intended.

Yes, that was a `bug' in our latex code. Thank you for your thorough reading.

Figure 10:

It appears your domain is very close to the equator since there is no geostrophic-type flow due to the inclusion of the Coriolis term. Instead it appears you have a down-gradient flow, without the effects of rotation. In the future, I'd suggest focusing on a domain that is shifted more from the equator so there are significant contributions from both the geostrophic and ageostrophic components in your flow field.

Thanks for your feedback. We will take this into consideration.

Title: Estimating the State of a Geophysical System with Sparse Observations : Time Delay Methods to Achieve Accurate Initial States for Prediction

Recommendation: Minor Revision General Comment I think the authors have overall well addressed my original concerns and have indeed substantially modified and improved the manuscript also in the light of the other Reviewer' comments. I have only a few additional minor remarks for the Authors to address before acceptance. The comments that follow refer to the last Authors' reply and the numbering is made accordingly.

Specific Comment

 Abstract. I understand that Ls is a lower boundary, so that synchronization is achieved when L ≥ Ls. If so please add this piece of information in the text since the name critical number does not fully explain that it is also a minimum number. Also, please explain what the authors mean by synchronization here; a word explanation will suffice.

Ls is no longer referenced in the abstract, but we have added the term `lower bound' to the fifth paragraph where it is introduced.

This lower bound was termed the critical number of measurements $L_s$ required to synchronize the model with the data. Synchronization occurs when the error between the model state estimate and the data drops below a prescribed threshold, which is typically below the magnitude of the observation noise.

On point (4). The paper is still wrongly cited, as Pazo (2016) instead of Pazo et al. (2016), in line 7 page 6.
Sorry, this is fixed now.

On point (18). I could not find where these statements are given in the current version of the manuscript. Maybe they have been removed, which is fine with me. Please clarify this and point the exact portion of the text. If the authors refer to the sentences between lines 4-7 at page 9, those are fine with me but I do not straightforwardly see how they relate to the statements originally under discussion.
To the end of section 5.2 we have added
However, we have not developed a systematic way of choosing these values, and it is known from classical results on synchronization that the optimal choice depends on the number and distribution of observations.
We have removed the statement regarding the model resolution and the observation network.

On point (19). Here as well, I have problems to identify how and where in the text the authors have introduced modifications, if wished, to address my points. For instance, the current Section 5.1 does not seem to be related to the original Section 5.1, instead this seems to be the current Section 5.4. This makes me difficult to judge the changes.
Yes, the noise results were moved to their own section, which is now 5.4. The section was rewritten to make it more clear how noise is added to the data.

Regarding points (1) and (2), I could not find the equations in the current version of the manuscript. Are they still shown somewhere?

These equations were removed, they were confusing.

As for my point (3), I referred to an apparent inconsistency between the type of observations you used. From everywhere in the text is clear you are working with height measurements, but from the equation in the original Section 5.1, it seemed you also create synthetic observations of the velocity. This issue does not seem to be present anymore in the current version (Section 5.4).

Yes, noise is only added to the heights. There are no observations made on velocity. The misleading equations were removed.

Please indicate where in the text are placed the corrections mentioned in points (4) and (5).

References to C_height and C_data have been removed and the text in section 5.4 is now consistent with Fig 7.

On point (23). Please provide information on which actions, if any, you have taken to address this remark. By looking at the current version of the conclusion, it seems that all discussion on equivalence with the smoothers has been removed. Has it been placed elsewhere or modified somehow? Please clarify.

The discussion on the connection with 4DVar was revised considerably and moved to the end of Section 2. We plan to give a more detailed comparison between the two methods in a future paper.